# Global epistasis emerges from a generic model of a complex trait

Gautam Reddy[1]*, Michael M Desai[1,2,3,4]*

[1]NSF-Simons Center for Mathematical and Statistical Analysis of Biology, Harvard University, Cambridge, United States; [2]Department of Organismic and Evolutionary Biology, Harvard University, Cambridge, United States; [3]Quantitative Biology Initiative, Harvard University, Cambridge, United States; [4]Department of Physics, Harvard University, Cambridge, United States

**Abstract** Epistasis between mutations can make adaptation contingent on evolutionary history. Yet despite widespread 'microscopic' epistasis between the mutations involved, microbial evolution experiments show consistent patterns of fitness increase between replicate lines. Recent work shows that this consistency is driven in part by global patterns of diminishing-returns and increasing-costs epistasis, which make mutations systematically less beneficial (or more deleterious) on fitter genetic backgrounds. However, the origin of this 'global' epistasis remains unknown. Here, we show that diminishing-returns and increasing-costs epistasis emerge generically as a consequence of pervasive microscopic epistasis. Our model predicts a specific quantitative relationship between the magnitude of global epistasis and the stochastic effects of microscopic epistasis, which we confirm by reanalyzing existing data. We further show that the distribution of fitness effects takes on a universal form when epistasis is widespread and introduce a novel fitness landscape model to show how phenotypic evolution can be repeatable despite sequence-level stochasticity.

**\*For correspondence:**
gautam_nallamala@fas.harvard.edu (GR);
mdesai@oeb.harvard.edu (MMD)

**Competing interests:** The authors declare that no competing interests exist.

## Introduction

Despite the idiosyncrasies of epistasis, a number of laboratory microbial evolution experiments show systematic patterns of convergent phenotypic evolution and declining adaptability. A striking example is provided by the *Escherichia coli* long-term evolution experiment (LTEE) (*Figure 1a*): 12 replicate populations that adapt in parallel show remarkably similar trajectories of fitness increase over time (*Wiser et al., 2013*; *Lenski et al., 2015*), despite stochasticity in the identity of fixed mutations and the underlying dynamics of molecular evolution (*Tenaillon et al., 2016*; *Good et al., 2017*). Similar consistent patterns of fitness evolution characterized by declining adaptability over time have also been observed in parallel yeast populations evolved from different genetic backgrounds and initial fitnesses (*Kryazhimskiy et al., 2014*; *Figure 1b*) and in other organisms (*Elena and Lenski, 2003*; *Perfeito et al., 2014*; *Wünsche et al., 2017*; *Sanjuán et al., 2005*; *Couce and Tenaillon, 2015*; *Jerison et al., 2017*; *Schenk et al., 2013*). Declining adaptability is thought to arise from diminishing-returns epistasis (*Khan et al., 2011*; *Chou et al., 2011*; *Kryazhimskiy et al., 2014*), where a global coupling induced by epistatic interactions systematically reduces the effect size of individual beneficial mutations on fitter backgrounds. Diminishing-returns manifests as a striking linear dependence of the fitness effect of a mutation on background fitness (*Figure 1c*). While diminishing-returns can be rationalized as the saturation of a trait close to a fitness peak, recent work shows a similar dependence on background fitness even for deleterious mutations, which become more costly on higher fitness backgrounds (*Johnson et al., 2019*). This suggests that fitter backgrounds are also less robust to deleterious effects (*Figure 1d*), a phenomenon that has been termed increasing-costs epistasis. The origin of the global coupling that results in these effects is unknown.

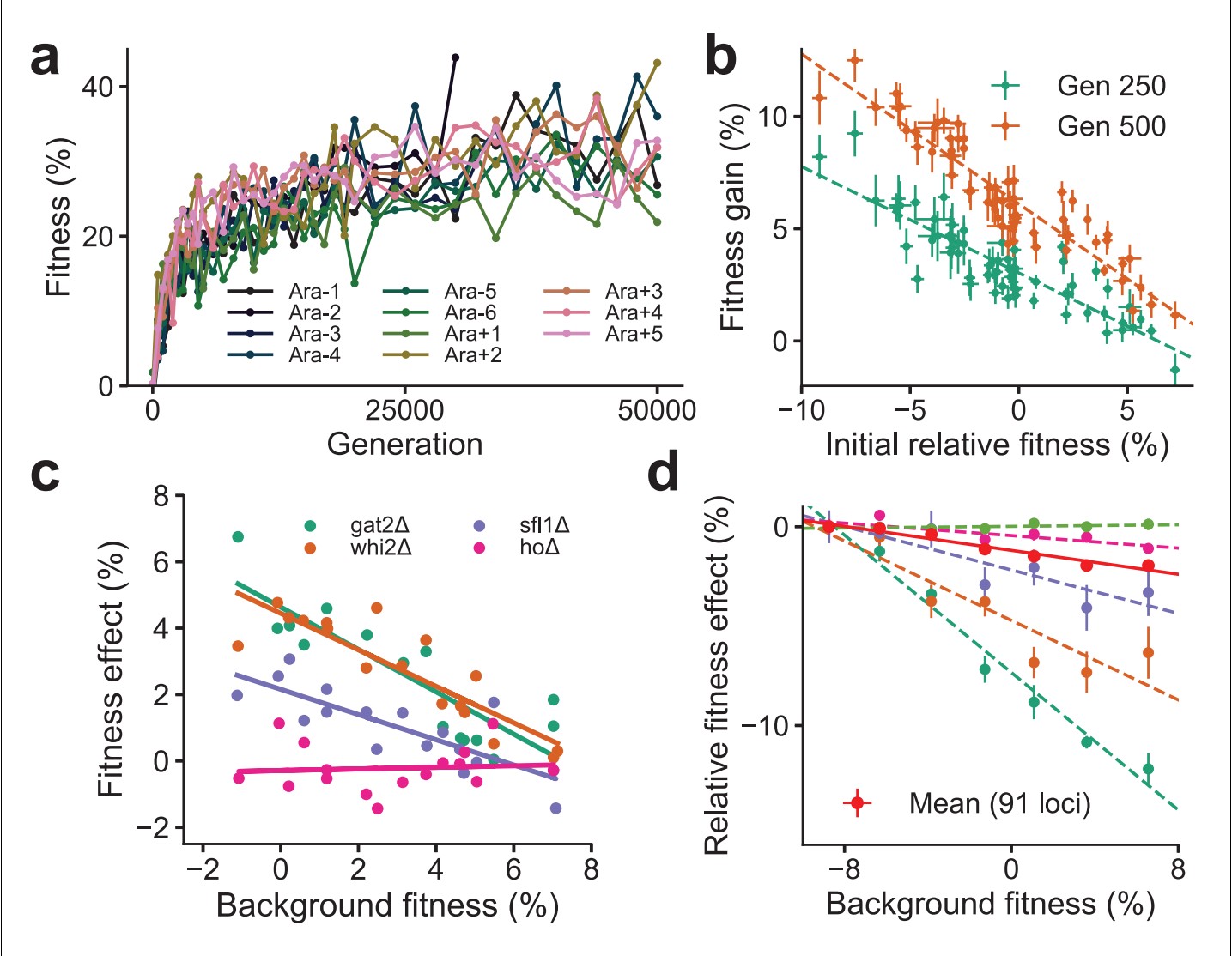

**Figure 1.** Declining adaptability and global epistasis in microbial evolution experiments. (**a**) Convergent phenotypic evolution in the *E. coli* long-term evolution experiment: the fitness relative to the common ancestor of 11 independently adapting populations over 50,000 generations is shown (data from *Wiser et al., 2013*). The 12th population, Ara + 6, has limited data and is not shown. (**b**) Yeast strains with lower initial fitness adapt faster (data from *Kryazhimskiy et al., 2014*). The fitness gain after 250 (green) and 500 (orange) generations of 640 independently adapting populations with 64 different founders and 10 replicates of each founder. Mean and SE are computed over replicates. (**c**) Diminishing returns of specific beneficial mutations on fitter backgrounds for three knocked out genes (green, orange, and purple) (data from *Kryazhimskiy et al., 2014*). Control in pink. (**d**) Increasing costs of specific deleterious mutations on fitter backgrounds (data from *Johnson et al., 2019*). The fitness effect relative to the least fit background for the mean over 91 mutations (in red) and 5 of the 91 mutations is shown. Linear fits for the five specific mutations and the mean using dashed and solid lines respectively are shown.

Put together, these empirical observations suggest that the contributions to the fitness effect, $s_i$, of a mutation at a locus $i$ in a given genetic background can be written as

$$s_i = s_{\mathrm{additive},i} + s_{\mathrm{genotype},i} - c_i y, \tag{1}$$

where $s_{\mathrm{additive},i}$ is the additive effect of the mutation, $s_{\mathrm{genotype},i}$ is its genotype-dependent epistatic contribution independent of the background fitness $y$ (i.e., idiosyncratic epistasis), and $c_i$ quantifies the magnitude of global epistasis for locus $i$. *Equation (1)* reflects the observation that the strength of global epistasis depends on the specific mutation and applies independently of whether its additive effect is deleterious (increasing-costs) or beneficial (diminishing-returns). Over the course of

adaptation in a fixed environment, global epistatic feedback on mutational effects can lead to a long-term decrease in adaptability. If this feedback dominates, *Equation (1)* suggests that the dependence of the fitness effect on evolutionary history is summarized entirely by the current fitness, and therefore results in predictable fitness evolution.

Here, we show that diminishing-returns and increasing-costs epistasis are a simple consequence of widespread epistasis (WE). This is consistent with recent work (*Lyons et al., 2020*) that proposes a similar argument to explain these phenomena. However, while the core idea is similar, we present here an alternative framework based on the Fourier analysis of fitness landscapes, which leads to new insights and quantitative predictions. In particular, our framework leads to novel predictions for the relationship between the magnitude of global epistasis and the stochastic effects of microscopic epistasis, which we confirm by reanalyzing existing data. Extending this framework, we further quantify how the distribution of fitness effects (DFEs) shifts as the organism adapts and how the fitness effect of a mutation depends on the sequence of mutations that have fixed over the course of adaptation (i.e., historical contingency). While specific historical relationships depend on the genetic architecture, we introduce a novel fitness landscape model with an intuitive architecture for which the entire history is summarized by the current fitness. Using this fitness landscape model, we investigate the long-term dynamics of adaptation and elucidate the architectural features that lead to predictable fitness evolution.

## Results

### Diminishing-returns and increasing-costs epistasis

We begin by examining the most general way to express the relationship between genotype and fitness (i.e., to describe the fitness landscape). A map between a quantitative trait (such as fitness), $y$, and the underlying genotype can be expressed as a sum of combinations of $\ell$ biallelic loci $x_1, x_2, \dots, x_\ell$ that take on values $x_i = \pm 1$ (*Hordijk and Stadler, 1998*; *Neher and Shraiman, 2011*; *Weinberger, 1991*; *Szendro et al., 2013*; *Weinreich et al., 2013*):

$$y = \bar{y} + \sum_i f_i x_i + \sum_{i>j} f_{ij} x_i x_j + \sum_{i>j>k} f_{ijk} x_i x_j x_k + \dots, \qquad (2)$$

where $\bar{y}$ is a constant that sets the overall scale of fitness. The symmetric convention $x_i = \pm 1$ for the two allelic variants is less often used than $x_i = 0, 1$, but it is an equivalent formulation, which we employ here because it will prove more convenient for our purposes (see *Poelwijk et al., 2016* for a discussion). The coefficients of terms linear in $x_i$ represent the additive contribution of each locus to the fitness (i.e., its fitness effect averaged across genotypes at all other loci), the higher-order terms quantify epistatic interactions of all orders, and $\bar{y}$ is the average fitness across all possible genotypes. Importantly, *Equation (2)* makes apparent the idiosyncrasies induced by epistasis: a mutation at a locus with $\ell$ interacting partners has an effect composed of $2^{\ell-1}$ contributions.

To explicitly compute the fitness effect of a mutation at locus $i$ on a particular genetic background, we simply flip the sign of $x_i$, keeping all other $x_j$ constant, and write down the difference in fitness that results. This fitness effect will generally involve a sum over a large number of terms involving the $f$'s in *Equation (2)*. While this may suggest that an analysis of fitness effects via *Equation (2)* is intractable, the analysis in fact simplifies considerably if the locus has a significant number of independent interactions that contribute to the fitness (i.e., provided that the number of independent, nonzero epistatic terms associated to the locus is large). In this case, we show that the fitness effects of individual mutations decrease linearly with background fitness and the fluctuations around this linear trend are normally distributed. In other words, widespread independent idiosyncratic epistatic interactions lead to the observed patterns of diminishing-returns and increasing-costs epistasis.

We present a derivation of this result in the SI. Here, we explain the key intuition using a heuristic argument. The argument is based on a simple idea: for a well-adapted organism ($y > \bar{y}$) with complex epistatic interactions, a mutation is more likely to disrupt rather than enhance fitness. To be quantitative, consider a highly simplified scenario where some number $N$ of the $f$'s in *Equation (2)* are $\pm 1$ at random and the others are 0. In this case, the fitness of a given genotype is a sum of $N_+$ and $N_-$ interactions that contribute positively and negatively to the trait, respectively, each with unit

magnitude, so that $y = \bar{y} + N_+ - N_-$. When positive and negative interactions balance, the organism is in a 'neutrally adapted' state ($y \approx \bar{y}$). By selecting for positive interactions, adaptation generates a bias so that $N_+ > N_-$ and $y > \bar{y}$. If locus $i$ involved in a fraction $v_i$ of all of $N = N_+ + N_-$ interactions is mutated, the effect of the mutation, on average, is to flip the sign of $N_+ v_i$ positive interactions and $N_- v_i$ negative interactions. The new fitness is then $y_i = y - 2N_+ v_i + 2N_- v_i = \bar{y} + (1 - 2v_i)(y - \bar{y})$ and thus $s_i = y_i - y = -2v_i(y - \bar{y})$. The negative linear relation between the background fitness, $y$, and the fitness effect of the mutation, $s_i$, is immediately apparent and emerges as a systematic trend simply due to a sampling bias towards positive interactions. Of course, while this relation is true on average, it is possible that locus $i$ affects more or less positive interactions due to sampling fluctuations. Provided only that $N$ is large and the interactions are independent, these fluctuations are approximately Gaussian with magnitude $\sqrt{N v_i (1 - v_i)}$.

This basic argument holds beyond the simple model with unit interactions. In the more general case, if the mutation is directed from $x_i = -1 \to +1$, we show in the SI that its fitness effect, $s_i$, on a background of fitness $y$ can be written as

$$s_i = \underbrace{2f_i(1 - \tilde{v}_i)}_{\text{additive}} - \underbrace{2\tilde{v}_i(y - \bar{y})}_{\text{global epistasis}} + \underbrace{\tilde{\epsilon}_i}_{\text{genotype}} , \qquad (3)$$

where,

$$\tilde{v}_i \equiv \frac{\left( \sum_{j \neq i} f_{ij}^2 + \sum_{j > k \neq i} f_{ijk}^2 + \dots \right) - \left( \sum_{j \neq i} f_j f_{ij} + \sum_{j > k \neq i} f_{jk} f_{ijk} + \dots \right)}{\sum_{j \neq i} (f_j - f_{ij})^2 + \sum_{j > k \neq i} (f_{jk} - f_{ijk})^2 + \dots}, \qquad (4)$$

and $\tilde{\epsilon}_i$ is a genotype and locus-dependent term that is distributed across genotypes with mean zero and variance expressed in terms of the $f$'s from *Equation (2)* (see SI for details). In the following equation and similar ones henceforth, a summation such as $\sum_{j > k \neq i} f_{ijk}^2$ is meant to denote a sum over pairs $j, k$, where each pair appears only once and no pair that includes index $i$ appears. Symmetry of the $f$'s w.r.t. interchanged indices is also assumed (e.g., $f_{ijk} = f_{jik}$). The numerator of $\tilde{v}_i$ in *Equation (4)* is proportional to the covariance of fitness effects and background fitness, and the denominator is the variance of background fitness across genotypes. A similar equation for the case $x_i = +1 \to -1$ can be derived. The choice of $+1 \to -1$ or $-1 \to +1$ is simply a matter of convention. If the convention is reversed, the coefficients of odd-order in *Equation (2)*, that is, $f_i, f_{ijk}, \dots$, should also switch signs. It can be easily checked that reversing the signs of these quantities in the expression for $\tilde{v}_i$ above leads to the expression for $\tilde{v}_i$ when $x_i = +1 \to -1$.

Note that in general $\tilde{v}_i$ is not guaranteed to be positive and $\tilde{\epsilon}_i$ is arbitrary and determined by the genotype-fitness map. However, consistent patterns emerge when locus $i$ has a large number of independent, nonzero epistatic terms and the additive effects $f_1, f_2, \dots$ of its interacting partners are not much larger than the epistatic terms (defined further below), which we call the WE limit. In the WE limit, $\tilde{\epsilon}_i$ is normally distributed across genotypes with variance proportional to $\tilde{v}_i(1 - \tilde{v}_i)$. This follows from the same reasoning as in our heuristic argument with unit interactions above (see SI for details). In addition, $\tilde{v}_i$ is typically positive, giving rise to a negative linear trend (i.e., diminishing-returns and increasing-costs). We can see this by taking the third- and higher-order terms in *Equation (4)* to be zero, in which case $\tilde{v}_i$ is positive if $\sum_{j \neq i} f_{ij}^2 > \sum_{j \neq i} f_j f_{ij}$. This will typically be true in the WE limit because we expect $\sum_{j \neq i} f_{ij}^2$ to scale with the number of interacting partners $\ell$, while each term in $\sum_{j \neq i} f_j f_{ij}$ can be positive or negative and thus the sum scales as $\sqrt{\ell}$ if the terms are independent. Thus when locus $i$ has a large number of interacting partners, $\tilde{v}_i$ is typically positive unless the magnitude of the additive terms ($a$) is much larger than the magnitude of the epistatic terms ($e$), $a \gg e\sqrt{\ell}$. This argument is easily extended to the case when the third- and higher-order terms are nonzero (see SI); the upshot is that the bias towards $\tilde{v}_i$ positive gets stronger with increasing epistasis.

The conditions for the WE limit are more likely to hold when the number of loci, $\ell$, that affect the trait is large. Therefore, we expect to generically observe patterns of diminishing-returns and increasing-costs epistasis for a complex trait involving many loci. Importantly, whether we observe a negative linear trend does not depend on the magnitude of a locus' epistatic interactions relative to its own additive effect, but rather relative to the additive effects of its interacting partners. If we are not in the WE limit, and instead the additive effects dominate (i.e., $a \gg e\sqrt{l}$), then *Equation (4)*

suggests that the slope of the linear trend can be either positive or negative. We will show further below that recent experimental data demonstrates that both scenarios can be relevant: some loci have $a \ll e\sqrt{l}$ while others have $a \gg e\sqrt{l}$, with the former creating a bias towards the observed negative linear trends that characterize diminishing-returns and increasing-costs epistasis.

We note that *Equation (3)* immediately leads to testable quantitative predictions: in the WE limit, the distribution of the residuals, $\tilde{\epsilon}_i$, obtained from regressing $s_i$ and $y$ is entirely determined by the slope of the regression, $-2\tilde{v}_i$. Specifically, we predict that these residuals (the deviations of individual genotype fitnesses from the overall diminishing-returns or increasing-costs trend) should be normally distributed with a variance proportional to $\tilde{v}_i(1 - \tilde{v}_i)$. However, this condition only applies if diminishing-returns arises from the WE limit. It does not hold if epistasis is negligible, if locus $i$ interacts significantly with only a few other dominant loci, or if the epistatic terms are interrelated (e.g., when global epistasis arises from a nonlinearity applied to an unobserved additive trait; *Starr and Thornton, 2016*; *Sailer and Harms, 2017*; *Otwinowski et al., 2018*). The latter case may still lead to a negative linear trend, but the statistics of the residuals will differ from *Equation (3)* (see SI for a discussion).

It is convenient to subsequently work with the symmetric version of *Equation (3)*, where the fitness effects of both $x_i = -1 \rightarrow +1$ and its reversion $x_i = +1 \rightarrow -1$ (whose fitness effect is negative of the former) are included in the regression against their respective background fitness. In this case, the additive term is averaged out, and we show (SI) that in the WE limit,

$$s_i = -2v_i(y - \bar{y}) + 2\sqrt{v_i(1 - v_i)}\,\eta_i, \tag{5}$$

where $\eta_i$ depends on the genetic background and the locus, and is normally distributed with zero mean and variance $V$, and

$$v_i \equiv \frac{V_i}{V} = \frac{f_i^2 + \sum_{j \neq i} f_{ij}^2 + \cdots}{\sum_k f_k^2 + \sum_{k>l} f_{kl}^2 + \cdots}. \tag{6}$$

Here, $V$ is the total genetic variance due to all loci (i.e., the variance in fitness across all possible genotypes) while $V_i$ is the contribution to the total variance by the $f$'s involving locus $i$. We therefore refer to $v_i$ as the *variance fraction* (VF) of locus $i$. We show further below that for certain fitness landscapes $v_i$ can also be interpreted as the fraction of pathways affected by a locus. For these reasons, we focus on $v_i$, which is half of the negative slope, rather than the slope. Note that the $v_i$'s do not sum to one unless there is no epistasis (with epistasis, $\sum_i v_i > 1$, reflecting the fact that the variance contributed by different loci overlap). While the directed mutation case discussed previously is the relevant one when presenting experimental data (e.g., *Figure 1c, d*), it is conceptually simpler to work with the symmetric case. These two cases coincide and $v_i \approx \tilde{v}_i$ in the WE limit if the additive effect of a locus is small (i.e., $f_i^2 \ll \sum_{j \neq i} f_{ij}^2 + \sum_{j>k \neq i} f_{ijk}^2 + \cdots$).

Our results show that the VF $v_i$ plays an important role. It determines the slope of the negative relationship between the fitness effect and background fitness. At the same time, it determines the magnitude of the idiosyncratic fluctuations away from this trend. We also note that this slope can be used to experimentally probe the contribution of a locus to the trait (i.e., its VF) taking into account *all* orders of epistasis, which circumvents the estimation of the individual $f$'s in *Equation (2)*. The theory additionally predicts that the slope obtained by regressing the sum of fitness effects of two mutations at loci $i, j$ against background fitness is proportional to $v_{ij} = v_i + v_j - 2e_{ij}$, where $e_{ij}$ quantifies the magnitude of epistatic interactions of all orders between $i$ and $j$ (SI).

Importantly, while the fitness effects of individual mutations (and hence the DFEs) may change over the course of evolution due to epistasis, the distribution of variance fractions (DVF) across loci, $P(v)$, is an invariant measure of the range of effect sizes available to the organism during adaptation. As we will see, this means that the DVF plays an important role in determining long-term adaptability.

## Numerical results and experimental tests

To illustrate our analytical results, we first demonstrate that the effects described above are reproduced in numerical simulations. To do so, we numerically generated a genotype-phenotype map of the form in *Equation (2)*, with $\ell = 400$ loci and an exponential DVF, $P(v) = \bar{v}^{-1} e^{-v/\bar{v}}$, where $\bar{v} = 0.02$

(Materials and methods). This DVF is shown in *Figure 2a*. Note that $\bar{v}\ell \gg 1$ corresponds to an epistatic landscape; $\bar{v}\ell = 8$ chosen here thus corresponds to a model within the WE limit (note that $\tilde{v}_i \approx v_i$ in this parameter range). Using this numerical landscape, we measured the fitness effect of mutations at 30 loci across 640 background genotypes with a range of fitnesses (*Figure 2b*). Our results recapitulate the predicted linear dependence on background fitness (*Figure 1c, d*), with a negative slope equal to twice the VF predicted from *Equation (5)*. We further simulated the evolution of randomly generated genotypes similar to the experimental procedure used in *Kryazhimskiy et al., 2014* (*Figure 2c*), finding that our results reproduce the patterns of declining adaptability observed in experiments (*Figure 1b*). Note that ~10 mutations are fixed during this simulated evolution; declining adaptability here is not due to a finite-sites effect.

As described previously, *Equation (5)* implies a proportional relationship between the magnitude of global epistasis (quantified by the slope of the relationship between the fitness effect of a mutation and the background fitness) and the magnitude of microscopic epistasis (quantified by the residual variance around this linear trend); see also *Figure 3a*. We verify this relationship in simulations (*Figure 2d*). We predict that the slope obtained by regressing the sum of fitness effects of two

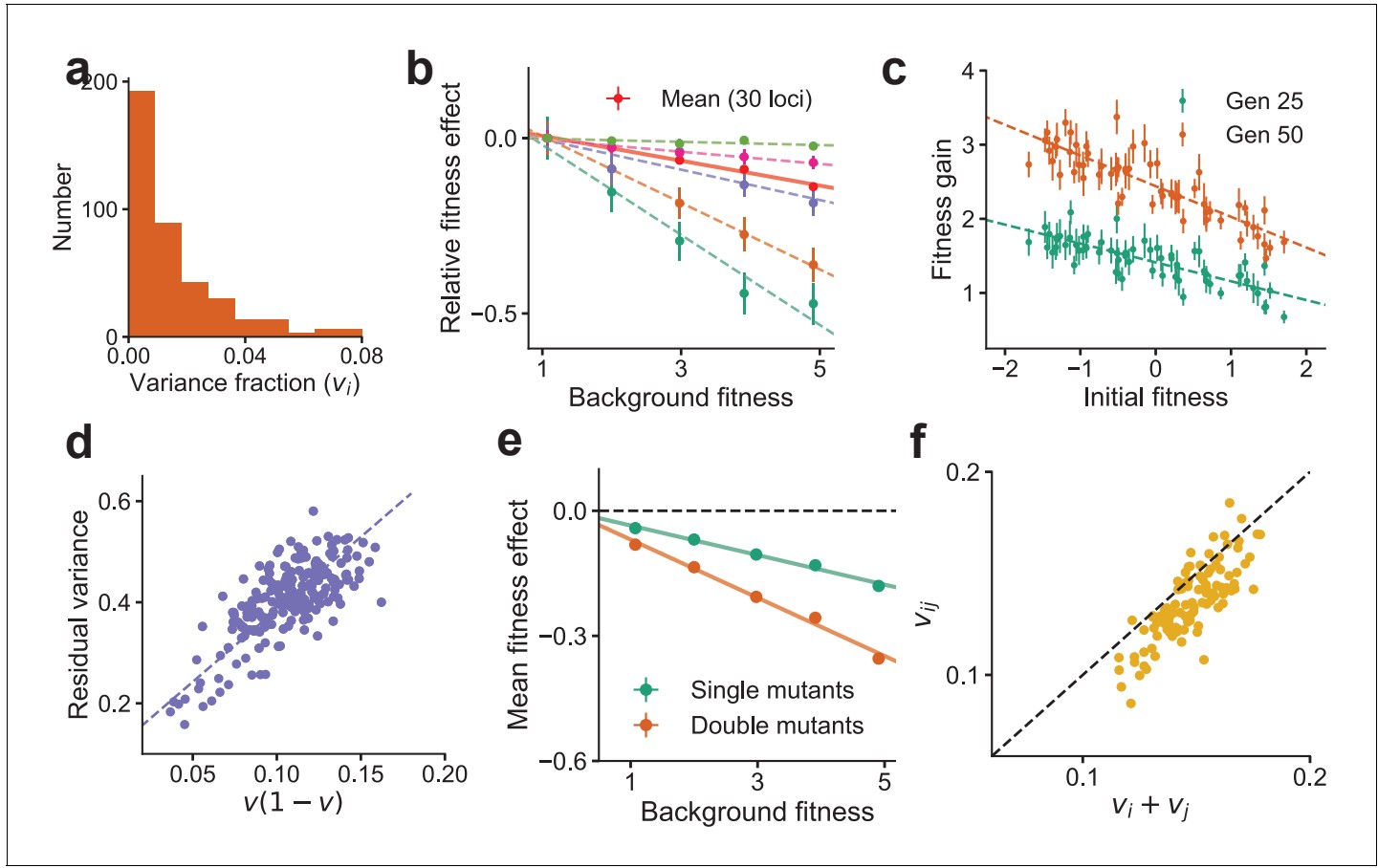

**Figure 2.** Global epistasis is recapitulated in a generic model of a complex trait and leads to testable predictions. (a) The distribution of variance fractions over 400 loci for the simulated genotype-phenotype map. (b) The predicted linear relationship between fitness effect (relative to the fitness effect on the least fit background) and background fitness for the mean over 30 randomly chosen loci (red, solid line) and five loci (dashed lines in colors) is recapitulated. The slope of the linear fit for each locus is proportional to its variance fraction, $v$ (slope = $-2v$). Mean and SE are over backgrounds of approximately equal fitness. See Materials and methods for more details. (c) The mean fitness gain after 25 (green) and 50 (orange) generations of simulated evolution of 768 independently adapting populations with 64 unique founders and 12 replicates each. Means and SEs are computed over the 12 replicates. Error bars are s.e.m. (d) The relationship predicted from theory between the residual variance from the linear fit for each locus and its slope is confirmed in simulations. (e) The mean fitness effect for single mutants at 30 loci and double mutants from all possible pairs of the 30 loci. The slope for the double mutants is predicted to be roughly twice that of single mutants. (f) The estimated variance fraction of a double mutant with mutations at two loci is predicted from theory and confirmed in simulations to be approximately the sum of the variance fractions for single mutations at the two loci. Sub-additivity is due to epistasis between the two loci. See Materials and methods for more details.

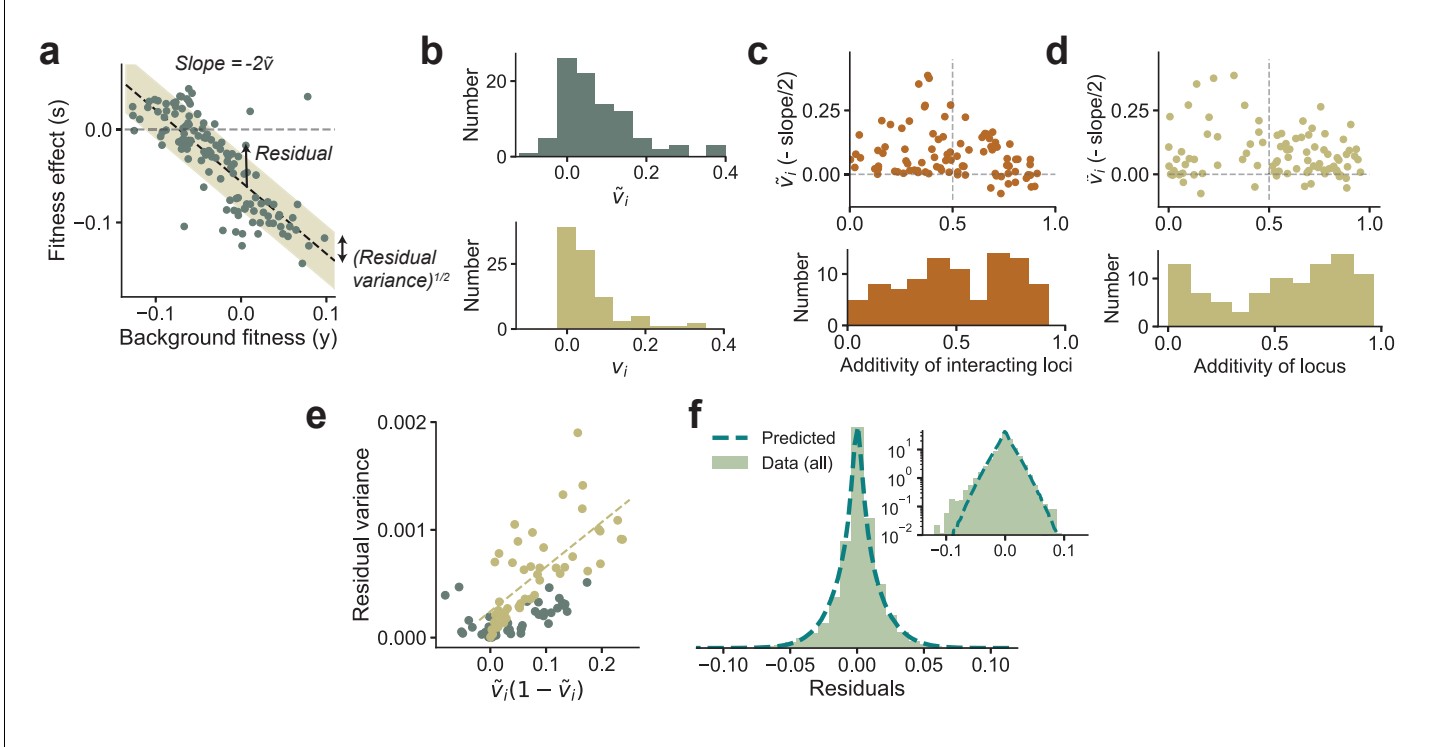

**Figure 3.** Experimental observations from *Johnson et al., 2019* are consistent with theoretical predictions. (a) The fitness effect of one of the 91 mutations from *Johnson et al., 2019* plotted against background fitness. (b) The distribution of the measured $\tilde{v}_i$ (negative one-half of the slope from a) and variance fractions $v_i$ for the 91 insertion mutations. (c, d) $\tilde{v}_i$ plotted against the additivity of interacting loci (AoIL) and the additivity of the mutated locus (see main text for definitions). The histograms are shown below the plots. The sign of the trend depends on the AoIL rather than the additivity of the mutated locus. (e) The measured variance of the residuals against the prediction $\tilde{v}_i(1 - \tilde{v}_i)$, shown here for the 91 mutations. The yellow circles correspond to the loci with AoIL $lt_{0.5}$. The best-fit line (yellow dashed line) to these loci has $R^2 = 0.50$ ($R^2 = 0.42$ for all points). (f) The shape of the distribution of residuals pooled from all 91 mutations aligns well with the prediction from *Equation (3)*. The variances of the two distributions are matched. Inset: same plot in log-linear scale. See Materials and methods for more details.

mutations at loci $i, j$ against background fitness is proportional to $v_{ij} = v_i + v_j - 2e_{ij}$. We further assume that $e_{ij} = O(\bar{v}^2)$ (specifically, $e_{ij} = v_i v_j$ for the genotype-phenotype map used for numerics). Since $v_i$ and $v_j$ are typically small for a complex trait, we expect near-additivity $v_{ij} \approx v_i + v_j$ and that any deviations are sub-additive, which is confirmed in simulations (*Figure 2e, f*).

While testing the latter prediction on double mutants requires further experiments, we can immediately test the relationship between the slope and the distribution of residuals from existing experimental data. To do so, we reanalyzed the data from *Johnson et al., 2019*, which measured the fitness effect of 91 insertion mutants on about 145 backgrounds. These background strains were obtained by crossing two yeast strains that differed by $\approx 40,000$ single-nucleotide polymorphisms (SNPs). Of these 40,000 loci, $\ell \approx 40$ have been identified as causal loci with currently available mapping resolution (*Bloom et al., 2015*). In *Figure 3*, we show the estimated $\tilde{v}_i$ (negative one-half of the slope of the best-fit line) and the VF $v_i$ for each of the 91 mutations. These mutations were selected after screening for nonzero effect, and thus the DVF is biased upwards. The mean VF is $\bar{v} \approx 0.06$. The wide range of $v_i$ observed in the data implies that the epistatic influence of loci varies greatly across loci, and we will show further below that this is crucial for maintaining a supply of beneficial mutations even when the organism is well-adapted to the environment.

Our theoretical results imply that we expect the linear relationship between background fitness and fitness effect to be negative if the additive effects of a locus' interacting partners are not much larger than the epistatic terms. Specifically, we define the additivity of interacting loci (AoIL) for locus $i$ as

$$\text{AoIL}(i) \equiv \frac{\left|\sum_{j\neq i} f_j f_{ij} + \sum_{j>k\neq i} f_{jk} f_{ijk} + \ldots\right|}{\left(\sum_{j\neq i} f_{ij}^2 + \sum_{j>k\neq i} f_{ijk}^2 + \ldots\right) + \left|\sum_{j\neq i} f_j f_{ij} + \sum_{j>k\neq i} f_{jk} f_{ijk} + \ldots\right|}, \tag{7}$$

which we show can be estimated from data (Materials and methods and SI). If the AoIL is less than half, *Equation (4)* implies that the linear trend is guaranteed to be negative. If instead the AoIL is greater than 0.5, the trend can be either positive or negative. The data shows a range of AoIL between 0 and 1 across loci. As predicted by our theory, we find that the loci with AoIL $lt_{0.5}$ always show negative trends and the ones with AoIL $gt_{0.5}$ show both negative and positive trends (*Figure 3c*). Importantly, the sign of the trend is determined by the AoIL and not by the additivity of the mutated locus, which we define as

$$\text{Additivity}(i) \equiv \frac{f_i^2}{f_i^2 + \sum_{j\neq i} f_{ij}^2 + \sum_{j>k\neq i} f_{ijk}^2 + \ldots}. \tag{8}$$

The additivity across loci also has a wide range. However, small additivity does not necessarily imply a negative trend (*Figure 3d*).

We next used the data from *Johnson et al., 2019* to analyze the relationship between the slope of the linear trend and the residual variance around this trend. We find that the experimental data confirms our theoretical prediction that the residual variance is proportional to $\tilde{v}_i(1 - \tilde{v}_i)$ if the AoIL is small (*Figure 3e*, $R^2 = 0.5$ for loci with AoIL ¡ 0.5 and $R^2 = 0.42$ for all loci). The Gaussian-distributed term in *Equation (3)* also predicts the shape of the distribution of the residuals given the VFs, which aligns well with the empirical distribution of the residuals (*Figure 3f*).

Together, these theoretical results and our reanalysis of experimental data show that linear patterns of global diminishing-returns and increasing-costs epistasis are a simple consequence of widespread epistatic interactions. The DVFs observed in data (*Figure 3b*) further imply that the epistatic influence of different loci on fitness can vary across a wide range. In what follows, we show that these two observations can be put together to make general predictions about the DFEes, and consequently the long-term dynamics of adaptation. The key ingredient that enables this analysis (including *Equation (5)*) is that in the WE limit fitness and fitness effects are jointly normal (with respect to a uniform distribution over all possible genotypes), which allows us to quantify complex dependencies between these variables in terms of pairwise covariances.

## The distribution of fitness effects

Long-term adaptation is determined by the DFEs of possible mutations and the stochastic dynamical processes that lead to fixation. While *Equation (5)* represents the distribution of the fitness effects of a specific mutation at locus $i$ over *all* genotypes in the population that have fitness $y$, we are instead interested in the DFE, where fitness effects are measured for all the mutations arising in the background of a *particular* genotype that has fitness $y$. For now we ignore the influence of evolutionary history on the DFE; we expand on that complication in the following section.

Examining the DFE over $\ell$ loci for a randomly chosen genotype of fitness $y$ can be thought of as sampling the fitness effects $s_1, s_2, \ldots, s_\ell$ from the conditional joint distribution $P(s_1, s_2, \ldots, s_\ell | y)$, which generally depends on epistasis. If the number of independent, nonzero epistatic terms is large, then $P(s_1, s_2, \ldots, s_\ell | y)$ is a multivariate normal distribution defined by the means and covariances of the $\ell + 1$ variables $y, s_1, s_2, \ldots, s_\ell$, which in turn can be computed in terms of the $f$'s from *Equation (2)*. In particular, the conditional means and covariances are $\text{Mean}_y(s_i) = -2v_i(y - \bar{y})$, $\text{Cov}_y(s_i, s_j) = 4V(e_{ij} - v_i v_j)$, where $e_{ij}$ is the epistatic VF between loci $i$ and $j$ and $e_{ii} = v_i$. This implies that the conditional correlation between fitness effects is $(e_{ij} - v_i v_j)/\sqrt{v_i v_j (1 - v_i)(1 - v_j)}$.

The DFE simplifies considerably if we make certain additional assumptions on the magnitude of epistatic interactions. If we assume the typical VF $\bar{v}$ is small (i.e., $\bar{v} \ll 1$) and also that $e_{ij}$ is $O(\bar{v}^2)$, then correlations are $O(\bar{v})$ and thus negligible. Then, in a particular sample $s_1, s_2, \ldots, s_\ell$, we can think of each $s_i$ as being drawn independently with mean $-2v_i(y - \bar{y})$ and variance $4v_i V$. To compute the DFE, $\rho(s|y)$, we first sample the VF from the DVF, $P(v)$, and then sample a Gaussian random variable with the aforementioned mean and variance. This leads to the DFE

$$\rho(s|y) = \int_0^1 dv (2\sqrt{vV})^{-1} P(v) \varphi\left(\frac{s + 2v(y-\bar{y})}{2\sqrt{vV}}\right),$$  (9)

where $\varphi$ is the standard normal pdf. Curiously, the correlations between $s_i$'s vanish when $e_{ij} = v_i v_j$, in which case the above equation is exact and the DFE is determined entirely by the DVF. Further below, we introduce a specific fitness landscape model for which this relation does hold. Diminishing-returns is naturally incorporated in *Equation (9)*: the mean of $s$ is $-2\bar{v}(y-\bar{y})$, that is, the DFE shifts progressively towards deleterious values with increasing fitness.

## Historical contingency in adaptive trajectories

A key unresolved question is the extent to which evolutionary history influences the DFE and the dynamics of adaptation (*Agarwala and Fisher, 2019*). That is, what does our theory say about historical contingency?

Suppose a clonal population of fitness $y_0$ accumulates $k$ successive mutations resulting in fitnesses $y_1, y_2, \ldots, y_k$. By virtue of arising on the same ancestral background, the fitness gain of a new mutation, $s_{k+1}$, is in general correlated with the full sequence of past fitnesses and the identity of the $k$ mutations through its epistatic interactions with them. Based on these correlations, we use well-known properties of conditional normal distributions (*Eaton, 1983*) to write

$$s_{k+1} = \sum_{i=0}^{k} w_{k+1,i} y_i + \epsilon,$$  (10)

where the weights $w_{k+1,i}$ depend on the VF ($v_{k+1}$) of the new mutation and its epistatic interactions with past mutations. Here, $\epsilon$ is the normally distributed residual that depends on the initial genotype and the weights (SI). *Equation (10)* is a generalization to a sequence of mutations of *Equation (5)*, which we can think of as the special case where $k = 0$.

To gain intuition, it is useful to first analyze *Equation (10)* when $k = 1$ (i.e., to compute the effect of a second mutation conditional on the first). In this case, we show in the SI that

$$s_2 \simeq -2v_2(y_1 - \bar{y}) + \frac{v_1 v_2 - e_{12}}{v_1} s_1 + \epsilon,$$  (11)

where $s_1 = y_1 - y_0$ is the fitness effect due to mutation 1. The first term on the right-hand side is the dependence on the fitness of the immediate ancestor, similar to the corresponding term in *Equation (5)*. The second term quantifies the influence of epistasis between loci 1 and 2 on $s_2$. When $e_{12} = v_1 v_2$, dependence on $s_1$ vanishes entirely and $s_2$ depends only on $y_1$. In contrast, if loci 1 and 2 do not interact, $e_{12} = 0$, and $s_2$ is, on average, larger *if* the mutation at 1 is beneficial compared to when it is deleterious. This has an intuitive interpretation: diminishing-returns applies to the overall fitness and the mechanism through which it acts is epistasis. However, if mutations 1 and 2 do not interact, then the increase in fitness corresponding to mutation 1 does not actually reduce the effect of mutation 2 (as expected by diminishing-returns) so the expected effect of mutation 2 is larger. This analysis suggests that during adaptation, since selection favors mutations with stronger fitness effects on the current background, a mutation that interacts less with previous mutations is more likely to be selected.

To identify the conditions under which history plays a minimal role, we would like to examine when $s_{k+1}$ depends only on the current fitness, $y_k$, and is independent of both the past fitnesses and idiosyncratic epistasis. If this were true, then *Equation (5)* would apply for new mutations that arise through the course of a single evolutionary path (i.e., the fitness effect of a new mutation is 'memoryless' and depends only on its VF and the current fitness). Surprisingly, such a condition does exist. We show that this occurs when the magnitude of epistatic interactions between the new mutation and the $k$ previous mutations, $e_{k+1,1:k}$, satisfies a specific relation: $e_{k+1,1:k} = v_{k+1} v_{1:k}$, where $v_{1:k}$ is the combined VF of the $k$ previous mutations (SI). In general, this condition is not satisfied, implying that there will be historical contingency that can be analyzed using the framework above. Remarkably, it turns out that a fitness landscape model for which the condition is satisfied does exist and arises from certain intuitive assumptions on the organization of biological pathways and cellular processes.

This fitness landscape model additionally serves as an example of a landscape where global epistasis can vary substantially across loci. We describe this model below.

## The connectedness model

We introduce the 'connectedness' model (CN model, for short). In this model, each locus $i$ is involved in a fraction $\mu_i$ of independent 'pathways,' where each pathway has epistatic interactions between all loci involved in that pathway (*Figure 4a*). The probability of an epistatic interaction between three loci $(i, j, k)$ is then proportional to $\mu_i \mu_j \mu_k$ since this is the probability that these loci are involved in the same pathway. When the number of loci $\ell$ is large, we show that in this model $v_i = \mu_i / (1 + \mu_i)$, and when $\ell$ is small, $v_i = \mu_i / \bar{\mu} \ell$, where $\bar{\mu}$ is the average over all loci (SI). The CN model therefore has a specific interpretation: the outsized contribution to the fitness from certain loci (large $v_i$) is due to their involvement in many different complex pathways (large $\mu_i$) and not from an unusually large perturbative effect on a few pathways. The distribution, $P(\mu)$, across loci determines the DVF.

Statistical fitness landscapes such as the NK model and the Rough Mt. Fuji model (*Neidhart et al., 2013*; *Agarwala and Fisher, 2019*; *Kauffman and Weinberger, 1989*; *Stadler and Happel, 1999*; *Aita et al., 2000*; *Altenberg, 1997*) are related to the CN model. Specifically, the CN model is a subclass of the broader class of generalized NK models (see *Hwang et al., 2018* for a review). However, often-studied fitness landscape models have one important difference that distinguishes them and gives qualitatively different dynamics of adaptation (shown further below): in contrast to the CN model, classical fitness landscapes are typically

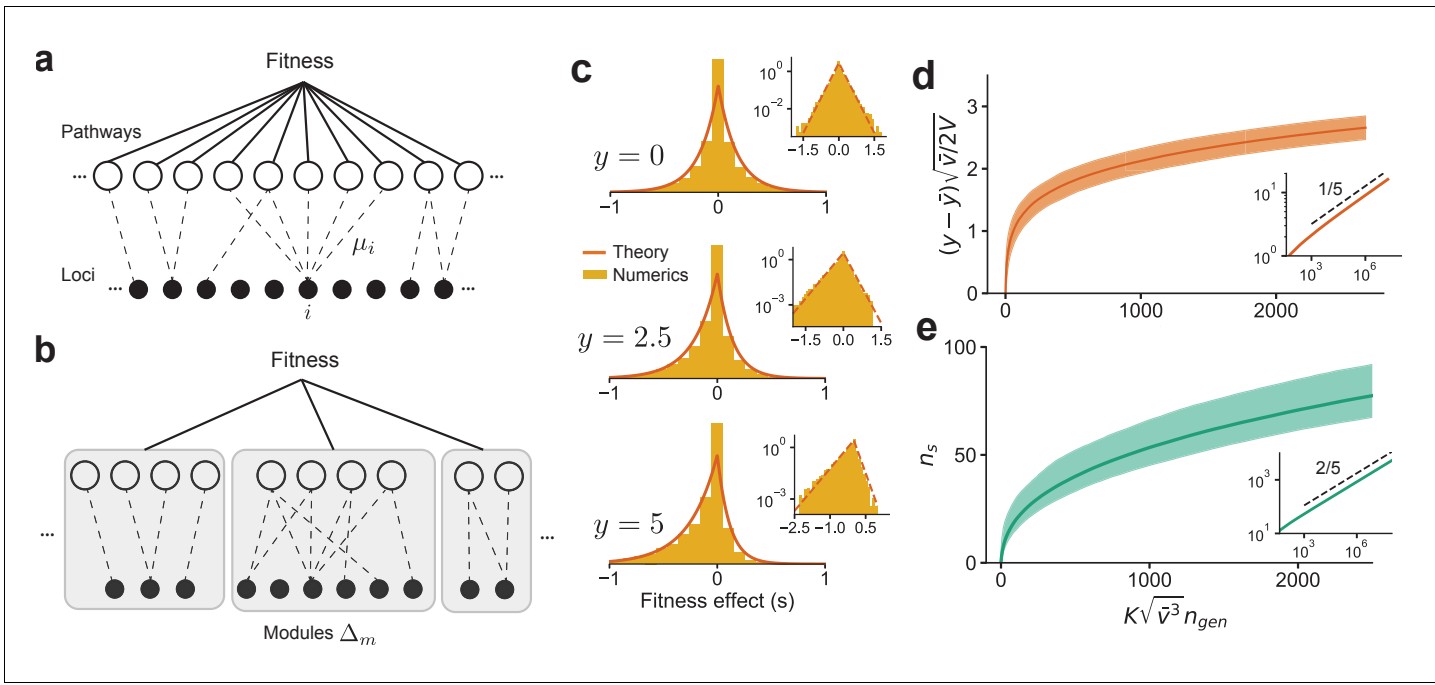

**Figure 4.** The distribution of fitness effect (DFE) and long-term adaptation dynamics predicted for the connectedness model. (a) Schematic of the connectedness (CN) model, where each locus is associated with a fraction μ of pathways that contribute to the organism's fitness. (b) An alternative model with modular organization, where sets of loci interact only within the pathways specific to a single module. (c) The DFE predicted from *Equation (14)* matches those obtained from simulated evolution of genotypes from the CN model. 128 randomly drawn genotypes (400 loci) with initial fitness $y$ close to zero are evolved to $y = 2.5$ and $y = 5$, and the DFE is measured across loci and genotypes. We chose $\bar{y} = 0$ and $V = 1$ so that $y$ represents adaptedness. Insets: same plots in log-linear scale. Note that the number of beneficial mutations acquired during the simulated evolution ($\sim 10$-$20$) is much less than the total number of loci (400). (d) For a neutrally adapted organism, the theory predicts quick adaptation to a well-adapted state beyond which the adaptation dynamics are independent of the specific details of the genotype-fitness map. Shown here is the mean adaptation curve predicted under strong-selection-weak-mutation (SSWM) assumptions, which leads to a power-law growth of fitness with exponent 1/5 in the well-adapted regime (inset). (e) The number of fixed beneficial mutations under SSWM, which grows as a power-law with exponent 2/5 in the well-adapted regime (inset). The shaded region is the 95 confidence interval around the mean for (c) and (d). See Materials and methods and SI for more details.

'regular.' That is, the VF of every locus is assumed to be the same (except the star neighborhood model, which has a bimodal DVF; *Hwang et al., 2018*).

The CN model is equivalent to a Gaussian fitness landscape with exponentially decaying correlations (SI). The CN model has tunable ruggedness, where the landscape transitions from additivity to maximal epistasis with increasing $\bar{\mu}$. Maximal epistasis corresponds to $\mu_i = 1$ (and hence $v_i = 1/2$) for all $i$. From *Equation (5)*, this implies that the new fitness after a mutation occurs is independent of the previous fitness, consistent with the expectation from a House-of-Cards (HoC) model (*Kauffman and Levin, 1987*) (where genotypes have uncorrelated fitness). Regular fitness landscape models with exponentially decaying correlations have memoryless fitness effects under the restrictive assumption that every locus is equivalent (*Agarwala and Fisher, 2019*). We show that the dynamics of adaptation of the more general CN model are also memoryless, that is, the condition detailed in the previous section holds true (SI). Yet, as we show below, the predicted dynamics for the CN model are very different to those from a regular fitness landscape model.

We emphasize that the well-connectedness assumed for the CN model is not a requirement for *Equation (5)* to hold. However, how diminishing-returns influences the long-term dynamics of adaptation depends on the specific genetic architecture and the corresponding fitness landscape. Consider, for example, an alternative model of genetic networks organized in a modular structure (*Figure 4b*). In this model, each locus is part of a single module and interacts epistatically with other loci in that module to determine the fitness of that module; overall fitness is then determined as a function of the module fitnesses. In this case, the variance contributed by a locus is due to its additive contribution and from epistasis between loci restricted to its module. While the argument for diminishing-returns still applies to the fitness as a whole, it follows from the same argument that diminishing-returns should also apply to each module separately. Consequently, the dynamics of adaptation for the modular model are different from the CN model. For simplicity, we analyze the dynamics of adaptation for the CN model and postpone a discussion of how the dynamics differ for different models to subsequent work.

## The dynamics of adaptation

We now examine the DFE that follows from *Equation (5)* and what that implies for long-term adaptation under the conditions for memoryless fitness effects. We henceforth assume a large number of loci with sparse epistasis (though the total number of nonzero epistatic terms is still large). This implies that $\ell \gg 1$, $v_i \ll 1$ and $\bar{v}\ell \gg 1$; for simplicity, we also assume strong-selection-weak-mutation (SSWM) selection dynamics and $s \ll 1$, $Ns \gg 1$, where $s$ are fitness effects and $N$ is the population size. Under these conditions, a mutation sweeps and fixes in a population before another one arises. The probability of fixation of a beneficial mutation, $p_{\text{fix}}$, is then proportional to its fitness effect (*Haldane, 1927*).

It is convenient to rescale fitnesses based on the total variance in fitness across all possible genotypes by defining $z = V^{-1/2}(y - \bar{y})$, $\sigma = V^{-1/2}s$, $\nu = V^{-1/2}\eta$. Note that $\nu$ is normally distributed with zero mean and unit variance. Here, $z$ has an intuitive interpretation as the 'adaptedness' of the organism. When the organism is neutrally adapted ($|z| \ll 1$), positive and negative epistatic contributions to the fitness are balanced and diminishing-returns is negligible. Diminishing-returns is relevant when the organism is well-adapted ($z \gg 1$). Below, we give the intuition behind our analysis, which is presented in full detail in the SI.

In the neutrally adapted regime, the linear negative feedback in *Equation (5)* is negligible and the DFE is determined by the distribution of $\simeq v^{1/2}\nu$. Loci with large $v$ can lead to a DFE with a long tail. If $\bar{v}$ is the typical VF of a locus, the fitness increases as $z \sim n_s\bar{v}^{1/2}$, where $n_s$ is the number of substitutions. Since $\bar{v}$ is a measure of overall epistasis, this implies that epistasis speeds adaptation in the neutrally adapted regime by allowing access to more influential beneficial mutations.

Fitness increases until the effect of the negative feedback cannot be neglected. From *Equation (5)*, this happens when $\bar{v}z \sim \bar{v}^{1/2}\nu$ (i.e., when $z^2 \sim \bar{v}^{-1}$). Intuitively, fitness begins to plateau when its accumulated benefit from substitutions is comparable to the scale of the total genetic variance ($n_s\bar{v} \sim 1$) and further improvements are due to rare positive fluctuations. In this well-adapted regime, diminishing-returns and increasing-costs epistasis strongly constrain the availability of beneficial mutations, whose effects can be quantified in this model: for a mutation to have a fitness effect $\sigma$, we require from *Equation (5)* that $\nu \simeq \sigma/2v^{1/2} + v^{1/2}z$, which has probability $\sim e^{-\nu^2/2}$. Beneficial

effects of large $\sigma$ arise when $\nu$ has a large positive deviation. The most likely $\nu$ that leads to a particular $\sigma$ is when $\nu$ is smallest (i.e., at $\nu^* \simeq \sigma/2z$), in which case $\nu \simeq \sqrt{2\sigma z}$, yielding a tail probability $\sim e^{-\sigma z}$. Remarkably, the beneficial DFE in the well-adapted regime is quite generally an exponential distribution independent of the precise form of the DVF (unless it is singular). In particular, we show in the SI that for the DFE, $\rho(\sigma|z)$,

$$\frac{\rho(\sigma|z)}{\rho(-\sigma|z)} = e^{-\sigma z}, \tag{12}$$

which depends solely on the adaptedness of the organism. The exponential form arises because of the Gaussianity of $\nu$, but the argument can be easily extended to $\nu$ with non-Gaussian tails.

An exponential beneficial DFE has been previously proposed by *Orr, 2003* but arises here due to a qualitatively different argument. Orr's result instead follows from extreme value theory: suppose the fitness effects of $\ell$ loci ($\ell \gg 1$) are sampled from a DFE $\rho(\sigma)$ and $F(\sigma) \equiv \int_{-\infty}^{\sigma} \rho(\sigma')d\sigma'$. Then, the probability that a beneficial mutation has at least a certain effect size $\sigma$ is $P(\sigma_b \geq \sigma) = \frac{1-F(\sigma)}{1-F(0)} \approx \frac{\ln F(\sigma)}{\ln F(0)}$, where the latter approximation holds when beneficial mutations are rare (i.e., $1 - F(0)$ is small). A well-known result from extreme value theory (*Gumbel, 2004*; *Majumdar et al., 2020*) implies that for a large family of distributions $\rho(\sigma)$ and for $\ell \gg 1$, we have $-\ell \ln F(\sigma) \propto e^{-k\sigma}$ (for some constant $k$) and therefore $P(\sigma_b \geq \sigma) = e^{-k\sigma}$. This argument is consistent with our results, but does not yield the dependence of $k$ on adaptedness and the rate of beneficial mutations without additional information about $\rho(\sigma)$.

Under SSWM assumptions, from *Equation (12)*, the typical effect size of a fixed mutation is $\sigma_{\text{fix}} \sim z^{-1}$, which typically has a VF,

$$\nu_{\text{fix}}^* \simeq \sigma_{\text{fix}}/2z \sim 1/2z^2. \tag{13}$$

The above relation makes precise the effects of increasing-costs epistasis on adaptation. As adaptation proceeds, the delicate balance of high fitness configurations constrains fixed beneficial mutations to have *moderate* VFs. A mutation of small VF is likely to confer small benefit and is lost to genetic drift, while one with a large VF is more likely to disrupt an established high fitness configuration.

This intuition is not captured in regular fitness landscape models, which assume statistically equivalent loci, that is, $\nu_i = \bar{\nu}$ for all $i$ and $P(\nu) = \delta(\nu - \bar{\nu})$ is singular. From *Equation (9)*, we see that this leads to a Gaussian DFE whose mean decreases linearly with increasing fitness, in contrast to the exponential DFE in our theory. The key difference is the lack of loci with intermediate effect, which drive adaptation in the well-adapted regime. As a consequence, the rate of beneficial mutations declines exponentially ($U_b \sim e^{-\bar{\nu}z^2/2}$) and the fitness thus sharply plateaus at $z \sim \bar{\nu}^{-1/2}$. In contrast, our theory predicts a much slower depletion of beneficial mutations, $U_b \sim z^{-2}$ (SI). The rate of adaptation is $dz/dt \sim U_b p_{\text{fix}} \sigma_{\text{fix}} \sim z^{-4}$ (since $p_{\text{fix}} \sim \sigma_{\text{fix}}$), which leads to a slow but steady power-law gain in fitness, $z \sim t^{1/5}$. The rate of fixation of beneficial mutations is $dn_s/dt \sim U_b p_{\text{fix}} \sim z^{-3} \sim t^{-3/5}$, which gives $n_s \sim t^{2/5}$.

We verify our analytical results using numerics. As before, we generated a genotype-phenotype map using the CN model with an exponential DVF, $P(\nu) = \bar{\nu}^{-1} e^{-\nu/\bar{\nu}}$ and $\ell = 400$ loci. The DFE can be calculated exactly by plugging in this $P(\nu)$ in *Equation (9)*:

$$\rho(\sigma|z) = \frac{\bar{\nu}^{-1}}{2\sqrt{2\bar{\nu}^{-1}+z^2}} e^{-\sigma z/2 - |\sigma|\sqrt{2\bar{\nu}^{-1}+z^2}/2}. \tag{14}$$

We simulated the evolution of randomly generated genotypes from $z = 0$ to $z = 2.5$ and $z = 5$, and the DFE across all loci was measured (we chose $\bar{y} = 0, V = 1$ so that $y = z, s = \sigma$). The theoretical prediction for the DFE, *Equation (14)*, closely aligns with the numerical results (*Figure 4c*).

Due to computational constraints, it is difficult to simulate evolution deep into the well-adapted regime. To compute the shape of adaptive trajectories and their variability, we instead simulated SSWM dynamics using the DFE directly from *Equation (14)*, beginning from a neutrally adapted fitness ($z = 0$). Typical trajectories (*Figure 4d*) show rapid adaptation to the well-adapted regime beyond which the fitness grows slowly as $t^{1/5}$, as predicted from theory. The predictions for the number of fixed beneficial mutation are also recapitulated (*Figure 4e*).

## Discussion

Recent empirical studies have observed consistent patterns of diminishing-returns and increasing-costs epistasis. Our model gives a simple explanation for these observations. In particular, we showed that these patterns are generic consequences of widespread microscopic epistatic interactions. The intuition underlying this result is that a random mutation typically has a larger disruptive effect on the delicate balance of microscopic epistasis that underpins a fitter background. Our model predicts a quantitative relationship between the magnitudes of global epistasis (i.e., the negative slope of diminishing-returns and increasing-costs epistasis) and microscopic epistasis, which we confirmed using existing data (*Figure 3*).

A similar explanation for diminishing-returns and increasing-costs epistasis has been recently proposed by *Lyons et al., 2020*. While our core argument for diminishing-returns and increasing-costs epistasis is the same as in that work, our Fourier analysis framework dissects the features of the fitness landscape necessary to observe these phenomena in terms of experimentally measurable average effects (i.e., the $f$'s in *Equation (2)*). In particular, we show that the additivity of a locus' interacting partners critically determines whether the trend is negative or unbiased. In addition, the Fourier analysis framework yields predictions for the DFEs, the historical influence of past mutations on the fitness effect of a newly mutated site, and motivates the proposed 'connectedness' fitness landscape model. The analysis of experimental data presented in Lyons et al. complements the experimental data considered here, lending further empirical support for the prevalence of epistasis and its importance in determining long-term adaptability.

Our model leads to other experimentally testable predictions. The most direct and accessible test of the theory is to measure the fitness for all possible combinations of mutations at $\sim 10$–15 significant loci and compare (using *Equation (6)*) the magnitude of global epistasis to the measured fitness coefficients (the $f$'s). Additionally, we predict that the magnitude of global epistasis of a double mutant should be nearly the sum of magnitudes of the corresponding single mutants, and any deviations should be biased towards sub-additivity. Since the predictions involve measuring residual variance, experimental noise can be an important confounding factor.

The observation that diminishing-returns occurs as a 'regression to the mean' effect on certain fitness landscapes has been noted previously (*Draghi and Plotkin, 2013*; *Greene and Crona, 2014*). The theory developed here quantifies precisely when we should expect to observe these patterns. We emphasize that our key result, *Equation (5)*, is a general statistical relation that holds if epistasis is widespread, irrespective of the specific genetic architecture and the corresponding fitness landscape. Weak epistasis with many loci is sufficient to observe noticeable patterns of global epistasis. However, the argument fails if the contribution of a locus is purely additive or when epistasis is limited to one or a handful of other loci. In the latter case, we expect the fitness effect of a mutation to be dominated by the allelic states of its partner loci, and thus take on a few discrete values. A few examples from *Johnson et al., 2019* indeed exhibit this pattern (e.g., cases where the fitness effect of a specific mutation depends primarily on the allelic state at a single other locus).

We highlight a distinction between global epistasis discussed in this work and another form of global epistasis (also known as 'nonspecific' epistasis) typically used in protein evolution to describe nonspecific epistatic interactions due to a nearly additive trait transformed by a nonlinear function (*Starr and Thornton, 2016*; *Otwinowski et al., 2018*; *Otwinowski, 2018*; *Sailer and Harms, 2017*; *Husain and Murugan, 2020*). This nonlinear function creates systematic relationships between epistasis terms and breaks the condition of independent epistatic terms required for our arguments to apply. Specific nonlinearities such as an exponential function may indeed lead to a negative linear trend on average, but the structure of the residuals differs from the one in *Equation (5)* and observed in data.

A surprising empirical observation is that the negative linear relationship between fitness effect and ancestral fitness characteristic of global epistasis has different slopes for different loci. Our model identifies the negative slope as twice the fraction of variance contributed by a locus to the trait. To explain the wide range of VFs observed in data, we developed the CN model, a framework to think about the organization of cellular processes that can lead to loci of widely varying VFs. In the CN model, loci have a large VF due to their involvement in many different pathways rather than due to a large effect on a single pathway. The CN model can be viewed as a statistical fitness landscape where loci can have a range of VFs, specified by the DVFs. In the special case of every locus

having the same VF, the CN model corresponds to a fitness landscape with tunable ruggedness and exponentially decaying correlations.

Extending our framework to incorporate adaptation, we showed that the DFE depends only on the current fitness, rather than the entire evolutionary history, under the intuitive assumptions behind the CN model. The theory therefore gives a simple explanation for why phenotypic evolution can be predictable, even while the specific mutations that underlie this evolution are highly stochastic.

Our framework has an implicit notion of 'adaptedness' without referencing a Gaussian-shaped phenotypic optimum, often assumed in models of adaptation (e.g., Fisher's geometric model) (*Fisher, 1930*; *Orr, 2005*; *Martin and Lenormand, 2006*). Over the course of adaptation, the DFE shifts towards deleterious values, reflecting diminishing-returns, which naturally arises from our basic arguments. For a well-adapted organism, we show that the DFE for beneficial mutations takes on an exponential form and leads to universal adaptive dynamics. While an exponential DFE for beneficial mutations has been proposed previously based on extreme value theory (*Orr, 2003*), our result arises due to an entirely different argument: the tail of the beneficial DFE is determined by loci of intermediate size whose disruptive effect due to increasing-costs is small, yet whose effect size is large enough not to be lost due to genetic drift.

Our theory further predicts declining adaptability, with rapid adaptation in a neutrally adapted regime followed by much slower increases in fitness, resulting in power-law adaptive trajectories when the organism is well-adapted. This is consistent with observations from the *E. coli* LTEE (*Wiser et al., 2013*; *Lenski et al., 2015*). Our model predicts a quicker decline in the number of substitutions ($n_s \sim t^{2/5}$) compared to the near linear trend observed in the LTEE data (*Good et al., 2017*). However, the dynamics of fixation in the LTEE deviate strongly from SSWM assumptions. This may explain the discrepancy, although we note that existing theory has only analyzed the effects of clonal interference and other breakdowns in SSWM assumptions for a constant DFE and weak epistasis (*Good et al., 2012*; *Schiffels et al., 2011*). Further work will be required to understand how these effects interact with global epistasis. For example, we may expect that the effect of a highly beneficial mutation at a segregating locus is more likely to be attenuated due to interference from subsequent deleterious mutations, while a less-fit lineage has a larger pool of beneficial mutations and is thus more likely to 'leapfrog' over more-fit lineages.

## Materials and methods

The code and data to generate the figures are available at *Reddy, 2020*.

### Simulations

We use a fitness landscape model with $\ell$ loci to generate the genotype-fitness map. Each locus is assigned a sparsity $\mu$ from $P(\mu)$, which is an exponential distribution with mean $\bar{\mu}$. Each of $M$ independent pathways sample loci with each locus $i$ having probability $\mu_i$ of being selected to a pathway. We choose $\ell = 400, \bar{\mu} = 0.02, M = 500$ so that $\bar{\mu}\ell = 8$ ensures significant epistasis. All loci in a pathway interact with each other, where additive and higher-order coefficient terms of all orders were drawn independently from a standard normal distribution. The total fitness is the sum of contributions from the $M$ pathways. We normalize the coefficients so that the sum of squares of all coefficients is 1, that is, the total variance across genotypes is 1. The mean, $\bar{y}$, is close to zero from our sampling procedure. The above procedure is a simple and efficient way to generate epistatic terms to order $\sim 20$, beyond which the computational requirements are limited by the exponentially increasing demand. Note that the effects described in the paper were also observed with only pairwise and cubic epistatic terms.

The VFs shown in *Figure 2a* can be calculated numerically from the definition. From the theory, given our choice of $P(\mu)$, these should follow an exponential distribution with mean $\bar{v} \approx \bar{\mu}/(1 + \bar{\mu})$. There may be deviations since $M$ is finite whereas the calculations assume $M \to \infty$. To generate *Figure 2b*, in order to get a range of background fitnesses, we first sample 128 random genotypes. These have fitnesses close to zero; in order to obtain a range of fitness values, we simulated the evolution of these 128 genotypes up to $y = 1, 2, 3, 4, 5$ under SSWM assumptions to get $128 \times 5 = 640$ genotypes at roughly five fitness values. The fitness effect of applying a mutation (i.e., flipping its sign) is measured for 30 randomly chosen loci (which are kept fixed) over each of the 640 genotypes. This is shown for 5 of the 30 and for the mean over the 30 loci in *Figure 2b*.

To generate *Figure 2c*, we sampled 64 random genotypes and 12 replicates of each. The evolution of these 768 genotypes was simulated for a total of 50 generations with a mutation rate of 1 per generation. The mean fitness gain over the 12 replicates is plotted for each of the 64 founders against their initial fitness.

To generate *Figure 2d*, the residuals are measured using the same procedure as for the experimental data analysis described below for the initial 128 genotypes at $y \approx 0$ and the 30 loci with the largest VF.

Double mutants were created by mutating all pairs of the 30 randomly chosen loci on the 640 evolved genotypes. Their mean fitness effect was computed and plotted along with the mean fitness effect for single mutants, shown in *Figure 2e*. The VF of the pair of loci for the double mutant was estimated as before and compared to the sum of the estimated VFs of the corresponding single mutants. This is shown in *Figure 2f*.

To generate the plots in *Figure 4c*, we simulated the evolution of 128 randomly sampled genotypes to $y = 2.5$ and $y = 5$. The fitness effect of 200 randomly sampled loci was measured and the distribution is plotted.

## Analysis of the data from Johnson et al.

The data from *Johnson et al., 2019* consists of the fitness after the addition of 91 insertion mutations on each of 145 background genotypes. The fitness of a particular mutation at locus $i$ can be modeled as

$$y_i = -c_i y + b_i + \text{Residual}_i(g), \tag{15}$$

where $y_i, y$ are the mutant and background fitnesses, respectively, $c_i, b_i$ are constants for each locus, and the residual $\text{Residual}_i(g)$ depends on the background genotype $g$.

We estimate the VF $v_i = (1 - \hat{\rho}_i)/2$, where the Pearson correlation $\hat{\rho}_i = \text{Corr}(y_i \oplus y, y \oplus y_i)$, where the symbol $\oplus$ denotes that the mutant and background fitness datasets are concatenated. $\tilde{v}_i$ is estimated as the negative one-half of the slope of the best linear fit of $s_i = y_i - y$ and $y$. The residuals for each of the 145 genotypes for each of the 91 mutations are simply

$$\text{Residual}_i(g) = (y_i + c_i y) - \overline{(y_i + c_i y)}, \tag{16}$$

where the overline represents an average over the 145 genotypes, which is used as an estimate of the constant term and $c_i = 2\tilde{v}_i - 1$. In *Figure 3b*, we plot the distribution of estimated $v_i$ and $\tilde{v}_i$. In *Figure 3c*, we compute the AoIL for each locus using *Equation (7)*, which we show in the SI to be $|\text{Cov}(s_i, y_i + y)|/(|\text{Cov}(s_i, y_i + y)| + \text{Var}(s_i))$. In *Figure 3d*, we compute the additivity using *Equation (8)*. The additive effect is $f_i = \overline{(y_i - y)}/2$, and $\text{Var}(s_i)/4$ gives the sum of squares of the epistatic terms (SI). In *Figure 3e*, we compute the variance of the residuals across the 145 genotypes for each locus and plot it against the locus' estimated $\tilde{v}_i(1 - \tilde{v}_i)$. In *Figure 3f*, we plot the distribution of residuals over all genotypes and loci. The prediction is that in the WE limit the distribution of residuals is determined by $2\tilde{v}_i(1 - \tilde{v}_i)\eta$, where $\eta$ is a Gaussian random variable. We multiply $\sqrt{\tilde{v}_i(1 - \tilde{v}_i)}$ for each locus with 10,000 i.i.d. standard normal random variables, pool the resulting numbers for all loci, and plot the predicted distribution in *Figure 3f*. The distributions are variance-matched. While *Figure 3e* shows that the variance of the residuals aligns with the theoretical prediction of being proportional to slope, *Figure 3f* shows that the data is also consistent with the predicted Gaussianity of the background-genotype-dependent contribution.

## Acknowledgements

We thank Sergey Kryazhimskiy, Andrew Murray, Milo Johnson, and members of the Desai lab for comments on the manuscript. GR was supported by the NSF-Simons Center for Mathematical and Statistical Analysis of Biology at Harvard (award number #1764269) and the Harvard FAS Quantitative Biology Initiative. MMD acknowledges support from the Simons Foundation (grant 376196), NSF Grant PHY-1914916, and NIH grant R01GM104239.

## Additional information

### Funding

| Funder | Grant reference number | Author |
|---|---|---|
| Simons Foundation | NSF-Simons Center at Harvard #1764269 | Gautam Reddy |
| Simons Foundation | 376196 | Michael M Desai |
| National Science Foundation | PHY-1914916 | Michael M Desai |
| National Institutes of Health | R01GM104239 | Michael M Desai |

The funders had no role in study design, data collection and interpretation, or the decision to submit the work for publication.

### Author contributions

Gautam Reddy, Michael M Desai, Conceptualization, Formal analysis, Investigation, Methodology, Writing - original draft, Writing - review and editing

### Author ORCIDs

Gautam Reddy (iD) https://orcid.org/0000-0002-1276-9613
Michael M Desai (iD) http://orcid.org/0000-0002-9581-1150

### Decision letter and Author response

Decision letter https://doi.org/10.7554/eLife.64740.sa1
Author response https://doi.org/10.7554/eLife.64740.sa2

## Additional files

### Supplementary files

• Transparent reporting form

### Data availability

The code and data used to generate the figures are available at https://github.com/greddy992/global_epistasis (copy archived at https://archive.softwareheritage.org/swh:1:rev:ab20956034094e5789a5e0b74fb015d7f5341c31/).

The following previously published datasets were used:

| Author(s) | Year | Dataset title | Dataset URL | Database and Identifier |
|---|---|---|---|---|
| Wiser MJ, Ribeck N, Lenski RE | 2014 | Data from: Long-term dynamics of adaptation in asexual populations | https://datadryad.org/stash/dataset/doi:10.5061/dryad.0hc2m | Dryad, 10.5061/dryad.0hc2m |

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

## Appendix 1

### Diminishing-returns and increasing-costs epistasis in a model of a complex trait

In the main text, we propose that the fitness effect after a mutation at locus $i$ can be written as

$$s_i = -2v_i(y - \bar{y}) + 2\sqrt{v_i(1-v_i)}\eta_i, \quad 0 \leq v_i \leq 1. \tag{17}$$

Here, $\eta_i$ is a genotype and locus-dependent contribution, which is distributed as a mean-zero Gaussian with variance equal to the total genetic variance, $\bar{y}$ is a constant, and $v_i$ is the variance fraction, which is defined further below. *Equation (17)* corresponds to the symmetric case where the fitness effects of the mutations $x_i = -1 \rightarrow +1$ and $x_i = +1 \rightarrow -1$ are simultaneously regressed against their respective background fitness. The directed case when the mutation is specified to change from $x_i = -1 \rightarrow +1$ (or $x_i = +1 \rightarrow -1$) will be considered in the directed mutation section below.

We show that *Equation (17)* arises under certain conditions in a generic model of a complex trait with $\ell \gg 1$ biallelic loci. Essentially, we would like to compute the distribution of the new fitness, $y_i$, across genotypes after a mutation at locus $i$ given the current fitness $y$ and the set of all parameters $\Theta$ of the model (i.e., $P(y_i|y, \Theta)$, with $s_i = y_i - y$). Using the chain rule of probability, we can write

$$P(y_i|y, \Theta) = \sum_g P(y_i|g, \Theta)P(g|y, \Theta), \tag{18}$$

where the sum is over all possible genotypes. While $P(y_i|g, \Theta)$ is determined by the genotype-fitness map, $P(g|y, \Theta)$ is the crucial factor that gives weight only to the genotypes that yield the current fitness $y$. If the fitness is much larger than the mean fitness over all possible genotypes, *Equation (18)* implicitly ensures that weight is given to only those genotypic configurations that lead to such an unusually large fitness. We will analyze the case of a 'microcanonical ensemble' where every genotypic configuration that leads to a particular fitness is equally likely (with no linkage), that is, the prior $P(g)$ across genotypes is uniform (see the section 'Maximum entropy interpretation' for a discussion).

It is difficult to directly evaluate the sum in *Equation (18)*. In the following sections, we give a simple derivation but elaborated to highlight the key assumption that leads to *Equation (17)*. In short, the negative correlation between $s_i = y_i - y$ and $y$ implied by *Equation (17)* is a trivial consequence of $y_i$ and $y$ having the same distribution w.r.t. $P(g)$. $\eta_i$ is in general arbitrary and determined by the genotype-fitness map. However, $\eta_i$ is normally distributed if we make certain assumptions about the structure of epistasis in the genotype-fitness map. The emergence of the negative linear relationship for a directed mutation $x_i = -1 \rightarrow +1$ is subtler.

### Fourier representation of the fitness function

The fitness $y(g)$ for a genotype of length $\ell$, $g = \{x_1, x_2, \ldots, x_\ell\}$, where $x_i = \pm 1$, can be generally written as *Poelwijk et al., 2016*; *Neher and Shraiman, 2011*

$$y(g) = \bar{y} + \sum_i f_i x_i + \sum_{i>j} f_{ij} x_i x_j + \sum_{i>j>k} f_{ijk} x_i x_j x_k + \ldots \tag{19}$$

The symmetric choice of $x_i = \pm 1$ is chosen for mathematical convenience. In this form, the total variance contribution from the $q$th-order epistatic terms is $\sum_{i_1>i_2>\ldots i_q} f_{i_1 i_2 \ldots i_q}^2$, and the total genetic variance, defined as

$$V \equiv \frac{1}{2^\ell} \sum_g (y(g) - \bar{y})^2, \tag{20}$$

is the sum of variance contributions from orders $q = 1$ to $\ell$, that is, the sum of squares of all the $f$'s. The sum over the $2^\ell$ genotypes is an expectation value assuming all genotypes are equally likely; we will denote this expectation using an overline hereafter. We use this expectation value as a proxy for empirical averages over the 'background' genotypes in a population. With a sufficient number of background genotypes, the empirical average should converge to this expectation value.

The representation in *Equation (19)* is a Fourier representation of the fitness function on the $\ell$-dimensional hypercube and makes calculations much simpler. For instance, to get the terms from each order, we have

$$\bar{y} = \overline{y(g)}, \quad f_i = \overline{x_i y(g)}, \quad f_{ij} = \overline{x_i x_j y(g)}, \quad \dots \tag{21}$$

It is useful to define the variance contribution due to a particular locus $i$ as (symmetry w.r.t. interchanging indices is used for each term throughout)

$$V_i \equiv f_i^2 + \sum_{j \neq i} f_{ij}^2 + \sum_{k < j \neq i} f_{ijk}^2 + \dots. \tag{22}$$

We also define the *variance fraction* of locus $i$,

$$v_i \equiv \frac{V_i}{V}, \tag{23}$$

which plays a key role in the model.

## Derivation of *Equation (17)*

We would like to relate the fitness before and after locus $i$ is flipped (i.e., $x_i \to -x_i$), denoted by $y$ and $y_i$, respectively. We have from *Equation (19)*,

$$y - \bar{y} = y_{\bar{i}} + \xi_i, \tag{24}$$

$$y_i - \bar{y} = y_{\bar{i}} - \xi_i, \tag{25}$$

where $y_{\bar{i}}$ and $\xi_i$ respectively contain all the terms not containing and containing locus $i$, that is,

$$\xi_i = f_i x_i + \sum_{j \neq i} f_{ij} x_i x_j + \sum_{k < j \neq i} f_{ijk} x_i x_j x_k + \dots. \tag{26}$$

We have $y_{\bar{i}} = (y + y_i)/2 - \bar{y}$ and $\xi_i = (y - y_i)/2$. Since $y_{\bar{i}}$ and $\xi_i$ contain all the genotype dependence, we can write

$$P(y_i | y, \Theta) \propto P(y_i, y | \Theta) \tag{27}$$

$$= \int P(y_i, y | y_{\bar{i}}, \xi_i) P(y_{\bar{i}}, \xi_i | \Theta) dy_{\bar{i}} \xi_i \tag{28}$$

$$= P\left( y_{\bar{i}} = \frac{y + y_i}{2} - \bar{y}, \xi_i = \frac{y - y_i}{2} | \Theta \right). \tag{29}$$

From the properties of the Fourier representation in *Equation (21)*, it is easy to see that the means are $\overline{y_{\bar{i}}} = 0, \overline{\xi_i} = 0$, the variances are $\overline{y_{\bar{i}}^2} = V - V_i, \overline{\xi_i^2} = V_i$, and the covariance is $\overline{y_{\bar{i}} \xi_i} = 0$.

The calculations so far have been exact. We now make the key assumption that $y_{\bar{i}}$ and $\xi_i$ are both normal-distributed across genotypes. This assumption is similar to that of Fisher's infinitesimal model, where the distribution of trait values across strains for a complex trait is argued to be normal-distributed since the trait value is due to infinitesimal independent contributions from many loci. While $y_{\bar{i}}$ is easily seen to be normal-distributed for large $\ell$, an argument can be made for $\xi_i$ only if locus $i$ has a large number of independent, nonzero epistatic terms and the additive term $f_i$ is smaller in magnitude than the epistatic terms; specifically, we require that $f_i^2 \lesssim \sum_{j \neq i} f_{ij}^2 + \sum_{k < j \neq i} f_{ijk}^2 + \dots$. If instead $f_i^2 \gg \sum_{j \neq i} f_{ij}^2 + \sum_{k < j \neq i} f_{ijk}^2 + \dots$, then $\xi_i$ is bimodal, where the two modes correspond to $\xi_i \approx \pm f_i$ at $x_i = \pm 1$. For loci with pairwise and third-order epistasis, the number of pairwise and third-order epistatic terms scale $\propto \ell$ and $\propto \ell^2$, respectively, which justifies the normality assumption for large $\ell$ even if individual epistatic terms are smaller in magnitude than the additive terms.

Under these assumptions and since $y_{\bar{i}}$ and $\xi_i$ are linearly independent, we have

$$P(y_i | y, \Theta) \propto \varphi\left( \frac{y + y_i - 2\bar{y}}{2\sqrt{V - V_i}} \right) \varphi\left( \frac{y - y_i}{2\sqrt{V_i}} \right), \tag{30}$$

where $\varphi$ is the standard normal pdf. Therefore, $y_i$ is normal-distributed across genotypes and from the above equation can be written as

$$y_i - \bar{y} = (1 - 2v_i)(y - \bar{y}) + 2\sqrt{v_i(1 - v_i)}\eta_i, \tag{31}$$

where the VF $v_i = V_i/V$ was defined previously and $\overline{\eta_i} = 0, \overline{\eta_i^2} = V$. This leads to the form in **Equation (17)**,

$$s_i = -2v_i(y - \bar{y}) + 2\sqrt{v_i(1 - v_i)}\eta_i. \tag{32}$$

The above derivation was presented to clarify the basic assumptions. Simply computing the covariance between $y$ and $y_i$ in **Equations (24)** and **(25)**, we get $\overline{(y - \bar{y})(y_i - \bar{y})} = \overline{y_i^2} - \xi_i^2 = V - 2V_i$. The correlation is then $1 - 2v_i$. **Equation (32)** follows if additionally $y_i, y$ are jointly Gaussian, which is true if locus $i$ has many independent, nonzero epistatic terms.

## Directed mutation

Previously, we considered the symmetric flip $x_i \to -x_i$ and averaged over all $\ell$ loci including $i$. Here, we consider the case when the mutation is specified to change either from $x_i = -1 \to +1$ or $x_i = +1 \to -1$. In this case, we should average over all loci except $i$.

We consider $x_i = -1 \to +1$ (the opposite case is similar). The fitness before the mutation is

$$y_i^- = \bar{y} - f_i + \sum_{j \neq i}(f_j - f_{ij})x_j + \sum_{j>k \neq i}(f_{jk} - f_{ijk})x_j x_k + \ldots, \tag{33}$$

and the fitness after the mutation is

$$y_i^+ = \bar{y} + f_i + \sum_{j \neq i}(f_j + f_{ij})x_j + \sum_{j>k \neq i}(f_{jk} + f_{ijk})x_j x_k + \ldots, \tag{34}$$

so that

$$s_i = y_i^+ - y_i^- = 2f_i + 2\sum_{j \neq i}f_{ij}x_j + 2\sum_{j>k \neq i}f_{ijk}x_j x_k + \ldots. \tag{35}$$

The tilde and hat are used here to distinguish the fitness from the $y_i$ defined previously for the symmetric case corresponding to the flip $x_i \to -x_i$. We can easily compute averages over genotypes using the Fourier orthogonality relations. To compute the expectation value of products of two quantities, say $s_i$ and $y_i^-$, we simply compute a dot product where the components of the vectors are the coefficients of terms in the expansion above. For example, $\overline{s_i y_i^-}$ is $(\bar{y} - f_i)(2f_i) + \sum_{j \neq i}(f_j - f_{ij})(2f_{ij}) + \sum_{j>k \neq i}(f_{jk} - f_{ijk})(2f_{ijk}) + \ldots$. We have for the means, variances, and covariances,

$$\bar{s}_i = 2f_i, \tag{36}$$

$$\overline{y_i^-} = \bar{y} - f_i, \tag{37}$$

$$\overline{(s_i - \bar{s}_i)^2} = 4\sum_{j \neq i}f_{ij}^2 + 4\sum_{j>k \neq i}f_{ijk}^2 + \ldots, \tag{38}$$

$$\overline{(y_i^- - \overline{y_i^-})^2} = \sum_{j \neq i}(f_j - f_{ij})^2 + \sum_{j>k \neq i}(f_{jk} - f_{ijk})^2 + \ldots, \tag{39}$$

$$\overline{(s_i - \bar{s}_i)(y_i^- - \overline{y_i^-})} = 2\sum_{j \neq i}(f_j - f_{ij})f_{ij} + 2\sum_{j>k \neq i}(f_{jk} - f_{ijk})f_{ijk} + \ldots \tag{40}$$

$$= 2\left(\sum_{j\neq i} f_j f_{ij} + \sum_{j>k\neq i} f_{jk} f_{ijk} + \dots\right) - \frac{\overline{(s_i - \bar{s}_i)^2}}{2} \tag{41}$$

The slope when $s_i$ is regressed against $y_i^-$ is $\overline{(s_i - \bar{s}_i)(y_i^- - \overline{y_i^-})}/\overline{(y_i^- - \overline{y_i^-})^2}$. We can define a 'modified' VF $\tilde{v}_i$ as half the negative slope,

$$\tilde{v}_i \equiv \frac{\overline{(s_i - \bar{s}_i)^2}/4 - \left(\sum_{j\neq i} f_j f_{ij} + \sum_{j>k\neq i} f_{jk} f_{ijk} + \dots\right)}{\sum_{j\neq i}(f_j - f_{ij})^2 + \sum_{j>k\neq i}(f_{jk} - f_{ijk})^2 + \dots}, \tag{42}$$

$$= \frac{\left(\sum_{j\neq i} f_{ij}^2 + \sum_{j>k\neq i} f_{ijk}^2 + \dots\right) - \left(\sum_{j\neq i} f_j f_{ij} + \sum_{j>k\neq i} f_{jk} f_{ijk} + \dots\right)}{\sum_{j\neq i}(f_j - f_{ij})^2 + \sum_{j>k\neq i}(f_{jk} - f_{ijk})^2 + \dots}. \tag{43}$$

Writing the linear form based on this correlation, we get

$$s_i = 2f_i - 2\tilde{v}_i(y_i^- - \bar{y} + f_i) + K_i \eta_i, \tag{44}$$

where $\eta_i$ is again normally distributed (in the WE limit) with zero mean and variance $\overline{(y_i^- - \overline{y_i^-})^2}$ and $K_i^2 = \frac{\overline{(s_i - \bar{s}_i)^2}}{\overline{(y_i^- - \overline{y_i^-})^2}} - 4\tilde{v}_i^2$. Note that, unlike $v_i$, $\tilde{v}_i$ can be negative.

However, we argue that $\tilde{v}_i$ is typically positive in the WE limit, which leads to a negative linear trend. The second term in the numerator on the right-hand side of *Equation (42)* has the same number of terms as $\overline{(s_i - \bar{s}_i)^2}$, but these terms appear as products of Fourier coefficients that may have opposing signs. In particular, if $\sum_{j\neq i} f_{ij}^2 > \sum_{j\neq i} f_j f_{ij}$, $\sum_{j>k\neq i} f_{ijk}^2 > \sum_{j\neq i} f_{jk} f_{ijk}$, and so on, then $\tilde{v}_i$ is guaranteed to be positive. If we denote the typical magnitude of $q$th-order epistasis terms as $e_q$ ($e_1$ corresponds to additive effects), each of this relationships has the form $e_{q+1}^2 \ell > e_q e_{q+1} \sqrt{\ell}$ when $\ell \gg 1$, that is, $e_q < e_{q+1}\sqrt{\ell}$. If the number of loci is sufficiently large, then these relationships will hold even if the typical magnitude of individual higher-order epistasis terms is smaller than the lower-order terms. We therefore expect that the second term in the numerator on the right-hand side of *Equation (42)* is smaller than $\frac{\overline{(s_i - \bar{s}_i)^2}}{2}$ when $\ell \gg 1$. A similar argument can be made for the cross terms $f_j f_{ij}, f_{jk} f_{ijk}, \dots$ once the squares in the denominator of the right-hand side of *Equation (42)* are expanded.

When $\ell \gg 1$, we can then write

$$\tilde{v}_i \approx \frac{V_i - f_i^2}{V - f_i^2} = \frac{v_i - f_i^2/V}{1 - f_i^2/V} \approx v_i - f_i^2/V. \tag{45}$$

Further, in the WE limit, $K_i^2 \approx 4\tilde{v}_i - 4\tilde{v}_i^2$ so that the variance of $K_i \eta_i$ is $\propto \tilde{v}_i(1 - \tilde{v}_i)$.

To estimate the ratio of the magnitudes of the second and first terms in the numerator on the right-hand side of *Equation (42)* from data, we use the expression for the covariance,

$$\overline{(s_i - \bar{s}_i)(y_i^+ - \overline{y_i^+})} = 2\sum_{j\neq i}(f_j + f_{ij})f_{ij} + 2\sum_{j>k\neq i}(f_{jk} + f_{ijk})f_{ijk} + \dots, \tag{46}$$

to get

$$\frac{\left|\sum_{j\neq i} f_j f_{ij} + \sum_{j>k\neq i} f_{jk} f_{ijk} + \dots\right|}{\sum_{j\neq i} f_{ij}^2 + \sum_{j>k\neq i} f_{ijk}^2 + \dots} = \frac{|\mathrm{Cov}(s_i, y_i^-) + \mathrm{Cov}(s_i, y_i^+)|}{\mathrm{Var}(s_i)} = \frac{|\mathrm{Cov}(s_i, y_i^- + y_i^+)|}{\mathrm{Var}(s_i)} \tag{47}$$

## Comments on the result
### Fitness as a nonlinear function of an additive trait

The negative linear trend observed in data may arise due to the measured fitness being a nonlinear function of an unobserved additive trait. In this case, the epistasis terms are systematically related to

each other and the independence assumptions used to derive *Equation (32)* break down. In short, we show that specific nonlinearities can indeed lead to a negative linear trend, but the statistics of the residuals observed in data make this possibility unlikely.

Suppose the fitness is $y = \phi(u)$, where $\phi$ is a nonlinear function, $u = f_0 + \sum_i f_i x_i$ is the unobserved additive trait, $f_0$ is a constant, and $f_i$ are additive coefficients. For a linear trend, we require $s_i = \phi(u_i) - \phi(u) \propto \phi(u)$, where $u_i = u - 2f_i x_i$ for the flip $x_i \to -x_i$. For small $f_i$ relative to the other coefficients, we can Taylor expand $\phi(u_i)$ and show that we require $\phi(u) \propto e^u$ to get a linear trend. This nonlinearity creates a linear trend with slope $e^{-2f_i x_i} - 1$. For a negative linear trend, we require $2f_i x_i > 0$. However, even if this condition is true, the relation $s_i = (e^{-2f_i x_i} - 1)y$ is exact and there are no residuals.

To introduce residuals, suppose instead that $u = f_0 + \sum_i f_i x_i + h$, where $h$ is a HoC term, that is, it is an independent Gaussian random variable (mean 0, variance $\sigma_h^2$) across genotypes and repeatable across measurements. $h$ introduces epistasis in the unobserved trait. We have $s_i = (e^{-2f_i x_i + h_i - h} - 1)y$, where $h_i$ is the HoC term after the mutation, so that the average fitness effect conditional on $y$ is $\bar{s}_i = (e^{-2f_i x_i + \sigma_h^2} - 1)y$. The conditional variance of the residuals is $\overline{(s_i - \bar{s}_i)^2} = e^{-4f_i x_i + 2\sigma_h^2}(e^{2\sigma_h^2} - 1)y^2$. Note that the residual variance is no longer proportional to the slope and this variance increases as $y^2$, which are both inconsistent with the data.

## Maximum entropy interpretation

The expectation values are averages over all the genotypes assuming that every genotype of a particular fitness is equally likely, that is, the distribution over genotypes is uniform. This assumption is analogous to ensemble averages over a microcanonical ensemble in statistical physics, where one assumes that all the particle configurations that have a particular energy are equally likely. The experimental setting in *Johnson et al., 2019* is similar. The background genotypes are generated from a cross between two strains, which due to recombination makes each locus have equal probability of being one of the two alleles. Closely linked loci may be considered together as blocks. Some of the loci are partially linked, which may lead to deviations from the predictions. The expressions derived above can be easily extended to the case with different background genotype statistics.

The uniform distribution has an information-theoretic interpretation as the distribution that has the maximum entropy (MaxEnt) given no additional knowledge of how the genotype was generated. *Equation (32)* can therefore be viewed as the MaxEnt prediction of the fitness effect if locus $i$ is mutated conditioned on the current observed fitness $y$. A key idea that will be used throughout the paper is that when each locus $i$ has a significant number of independent, nonzero epistatic terms, the distribution of fitness and fitness effects is jointly normal with respect to the uniform prior over genotypes. From well-known properties of multivariate normal distributions, the MaxEnt predictions of unobserved variables are multilinear forms of the observed variables. For example, the MaxEnt prediction for $s_i$ given an observed sequence of past fitness is an autoregressive Gaussian process defined by the covariance between the unobserved and observed variables (see section 'History dependence' below).

## Varying a subset of loci

In addition, the sums in *Equation (19)* are over $\ell$ loci involved in a trait. In reality, we may vary a subset $P$ of all loci and take averages over only this subset. The derivation still follows through in this case; we can simply write the fitness in terms of effective parameters as

$$y(g) = \tilde{y} + \sum_{i \in P} \tilde{f}_i x_i + \sum_{i,j \in P} \tilde{f}_{ij} x_i x_j + \dots, \quad \text{where} \tag{48}$$

$$\tilde{y} = \bar{y} + \sum_{i \notin P} f_i x_i + \sum_{i,j \notin P} f_{ij} x_i x_j + \dots, \tag{49}$$

$$\tilde{f}_i = f_i + \sum_{j \notin P} f_{ij} x_j + \sum_{j,k \notin P} f_{ijk} x_j x_k + \dots, \quad \text{and so on.} \tag{50}$$

Here, we are abusing notation — it is to be assumed that a coefficient with a particular combination of indices appears only once in the sums. The results from the previous section still apply w.r.t. these new effective parameters. For instance, the total variance and the variance due to locus $i \in P$ are

$$\tilde{V} = \sum_{i \in P} \tilde{f}_i^2 + \sum_{i,j \in P} \tilde{f}_{ij}^2 + \sum_{i,j,k \in P} \tilde{f}_{ijk}^2 + \dots, \tag{51}$$

$$\tilde{V}_i = \tilde{f}_i^2 + \sum_{j \in P} \tilde{f}_{ij}^2 + \sum_{j,k \in P} \tilde{f}_{ijk}^2 + \dots, \tag{52}$$

and the effective VF is $\tilde{V}_i / \tilde{V}$.

## Relationship to the fluctuation-dissipation relation

*Equation (32)* is analogous to the fluctuation-dissipation theorem in statistical physics (**Kubo, 1966**), which relates the response of a thermodynamic system to a perturbation. The relationship between the magnitude of macroscopic epistasis (the slope in *Equation (32)*) and the variance due to the background genotypes is analogous to the relationship between viscous drag and Brownian motion (**Kubo, 1966**). For Brownian motion, the normality arises due to numerous independent collisions of a particle with neighboring particles. In our case, natural selection acts as an external perturbation that pushes the system away from equilibrium (here $\bar{y}$). Diminishing-returns naturally arises as the tendency of the system to revert to its entropically favored equilibrium state.

## Variance fraction of double mutants

Using the arguments from the previous section, it is easy to show that the variance of a double mutant at loci $i$ and $j$, $V_{ij}$, is necessarily sub-additive. In particular, we have

$$V_{ij} = V_i + V_j - 2I_{ij}, \quad \text{where} \tag{53}$$

$$I_{ij} = f_{ij}^2 + \sum_{k \neq i,j} f_{ijk}^2 + \dots \tag{54}$$

is the total *epistatic variance* between loci $i$ and $j$. The correlation between the new fitness after a double mutation and the previous fitness is $1 - 2V_{ij}/V \equiv 1 - 2v_{ij} = 1 - 2v_i - 2v_j + 2e_{ij}$, where $e_{ij} \equiv I_{ij}/V$ is the *epistatic variance fraction* between $i$ and $j$.

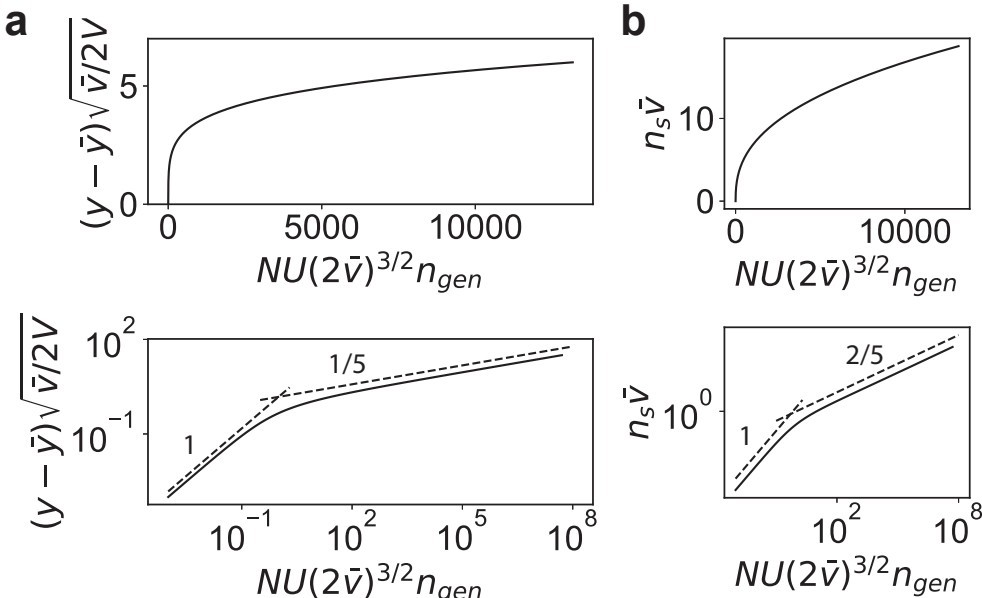

**Appendix 1—figure 1.** The mean fitness trajectory and the mean number of substitutions predicted by the model in the strong-selection-weak-mutation regime. (**a**) The scaled fitness vs. scaled time. Shown below is the fitness in log-scale to highlight the different scalings in the neutrally adapted and well-adapted regimes. The slopes of the dashed lines are shown. (**b**) The scaled number of substitutions vs. scaled time as in (**a**).

## Appendix 2

### Connectedness model

We introduce a 'connectedness' model (the CN model, for short), where each locus has a probability $\mu_i$ of being involved in any particular interaction. We can interpret $\mu_i$ as the fraction of independent pathways that involve locus $i$, where each pathway has epistatic interactions between all loci involved in that pathway. The number of independent pathways, $M$, is assumed to be very large. The probability of an epistatic interaction between, say loci $i, j, k$, is proportional to $\mu_i \mu_j \mu_k$ since this is the probability that these loci are involved in the same pathway. The magnitude of the interaction term is $f_{ijk}^2 \propto \mu_i \mu_j \mu_k$, where the proportionality is the magnitude of the perturbative effect of the mutations, which is assumed to be constant for all orders of interaction. We set this quantity to unity since it simply scales the fitness coefficients and does not affect subsequent results. The CN model leads to a specific interpretation (and hence its name): the outsized contribution to the variance from a particular locus is due to its involvement in many different complex pathways and not from an unusually large perturbative effect on a few pathways. For large $\ell$, the CN model is specified by the distribution, $P(\mu)$, across loci. In particular, given $P(\mu)$, we can calculate the total genetic variance, $V$, and the variance due to locus $i$, $V_i$. We define $\bar{\mu} \equiv \int_0^1 \mu P(\mu) d\mu$.

We calculate the *expected* total variance across statistical ensembles. Note that here the expectations are averages over ensembles where the parameters of the model are resampled, in contrast to the derivations presented in sections above, which were ensemble averages over equally likely genotypes. Since each pathway is independently sampled, expectations approximate the values in a single realization as $M \to \infty$. All expectations are denoted $\langle . \rangle$. We calculate the expected variance contribution from one pathway: since all pathways are statistically identical, the total variance from $M$ pathways is simply $M$ times the expected contribution from a single pathway. The contribution from the $q$th-order interaction between loci $i_1, i_2, \ldots, i_q$ is $\langle f_{i_1 i_2 \ldots i_q}^2 \rangle = \prod_{n=1}^q \mu_{i_n}$. The expected total variance is

$$V = \sum_i \langle f_i^2 \rangle + \sum_{i>j} \langle f_{ij}^2 \rangle + \sum_{i>j>k} \langle f_{ijk}^2 \rangle + \ldots, \tag{55}$$

$$= \sum_i \mu_i + \sum_{i>j} \mu_i \mu_j + \sum_{i>j>k} \mu_i \mu_j \mu_k + \ldots, \tag{56}$$

$$= \prod_{i=1}^\ell (1 + \mu_i) - 1 \tag{57}$$

The variance due to terms involving locus $i$ is

$$V_i = \langle f_i^2 \rangle + \sum_{j \neq i} \langle f_{ij}^2 \rangle + \sum_{j>k \neq i} \langle f_{ijk}^2 \rangle + \ldots, \tag{58}$$

$$= \mu_i + \mu_i \sum_{j \neq i} \mu_j + \mu_i \sum_{j>k \neq i} \mu_j \mu_k + \ldots, \tag{59}$$

$$= \mu_i \prod_{j \neq i} (1 + \mu_j). \tag{60}$$

Therefore, we have

$$v_i = V_i / V = \mu_i \frac{\prod_{j \neq i} (1 + \mu_j)}{\prod_j (1 + \mu_j) - 1}. \tag{61}$$

There are two qualitatively different regimes, $\bar{\mu}\ell \ll 1$ and $\bar{\mu}\ell \gg 1$. When $\bar{\mu}\ell \ll 1$, each pathway typically contains a single locus and should lead to an additive model. In this limit, we can write from *Equation (61)*,

$$v_i \approx \frac{\mu_i}{\ell\bar{\mu}}, \quad \bar{\mu}\ell \ll 1, \tag{62}$$

which is consistent with the expectation that $v_i \sim \ell^{-1}$ for an additive model. In the opposite limit, $\bar{\mu}\ell \gg 1$, we have

$$v_i \approx \frac{\mu_i}{1+\mu_i}, \quad \bar{\mu}\ell \gg 1. \tag{63}$$

The CN model therefore leads to an intuitive interpretation of the VF as being determined by the sparsity of interactions of a locus. *Equation (63)* yields an upper bound at $v_i = 0.5$, at $\mu_i = 1$. From *Equation (31)*, we see that this case is equivalent to the HoC model of maximal epistasis where the new fitness after a mutation is independent of the previous fitness. The DVF $P(v)$ is directly determined by $P(\mu)$; the CN model can therefore be used as a generative model to generate fitness landscapes with arbitrary DVFs.

We can further calculate the epistatic variance between two loci in the CN model. The total epistatic variance $I_{ij}$ between loci $i$ and $j$ is

$$I_{ij} = \langle f_{ij}^2 \rangle + \sum_{k \neq i,j} \langle f_{ijk}^2 \rangle + \dots \tag{64}$$

$$= \mu_i \mu_j \prod_{k \neq i,j} (1 + \mu_k). \tag{65}$$

In the limit $\bar{\mu}\ell \gg 1$, the epistatic VF after dividing by $V$ is then simply $e_{ij} = \mu_i \mu_j/(1+\mu_i)(1+\mu_j) = v_i v_j$. Using *Equation (53)*, we have

$$v_{ij} = v_i + v_j - 2v_i v_j. \tag{66}$$

If $v_i$'s are small, the CN model predicts near-additivity between the effects of two loci. This is not inconsistent with the strong epistasis assumption implicit in the limit $\bar{\mu}\ell \gg 1$: though the total contribution of epistatic interactions to the genetic variance may be large, the epistatic variance between two specific loci can still be negligible. This is because the majority of epistatic variance is due to the combinatorially large number of higher-order epistatic terms whose individual effects themselves can be weak.

## Relationship to statistical fitness landscapes

Statistical fitness landscapes such as the NK model and the Rough Mt. Fuji model are closely related to the CN model described above. The CN model falls under the broad class of generalized NK models (*Hwang et al., 2018*). In generalized NK models, epistasis is due to modules (or 'pathways') of $K$ loci that interact epistatically with each other. The different NK models differ in how the loci are assigned to the modules and the interaction structure within the module. In the CN model, each locus has a locus-specific probability of being part of any module and the interaction structure within a module is all-to-all. The locus-specific probability gives rise to a highly non-regular model, that is, loci can have a wide range of contributions to the total variance. This feature gives rise to qualitatively different adaptation properties. We will show this further below in addition to showing that the CN model has a special memoryless property.

Well-studied fitness landscapes such as Kauffman's NK model and the Rough Mt. Fuji landscape are regular, that is, every locus contributes equally to the variance. In other words, the DFE, which is determined by the DVF in our picture, instead comes from a single constant value $v = \bar{v}$. The fluctuations of the fitness effects are solely due to the genotype-dependent term $\eta$, which is a Gaussian. In a later section, we show that the DFE in this case corresponds to a Gaussian with mean $-2\bar{v}(y - \bar{y})$. As adaptation proceeds and $y$ increases, the DFE shifts to the deleterious side but retains its Gaussian shape. The adaptation properties that result from this DFE are quite different from those arising from our theory. Further, while the regular fitness landscape picture may lead to a good approximation of our results when the number of mutations is large, it does not capture the different magnitudes of diminishing-returns for different loci observed in experiments.

## The landscape of the regular CN model

The CN model for the special case of homogeneous loci, $\mu_i = \bar{\mu}$ for all $i$, is similar to the NK model but with one major difference: the number of loci in each pathway in the NK model is fixed at $K$ loci, whereas in the CN model the typical pathway size is controlled by a continuous parameter $\bar{\mu}$. This introduces contributions at every epistatic order while effectively imposing sparsity on the contributions from higher-order interactions. We now show that the regular CN model has tunable ruggedness, that is, it transitions from an additive model to a model with maximal epistasis with increasing $\bar{\mu}$ and has exponentially decaying correlations for $\bar{\mu}\ell \gg 1$.

The regular fitness landscapes often discussed are stationary Gaussian processes on a $\ell$-dimensional hypercube. The regular CN model (i.e., $\mu_i = \bar{\mu}$ for all $i$) also falls into this class as $M \to \infty$. The key quantity that defines such a fitness landscape is the covariance function between the fitnesses of genotypes $g$ and $g'$, $C(g, g') = C(d)$, where $d \equiv |g - g'|$ is the Hamming distance between two genotypes. We now compute the covariance $C(d)$ for the regular CN model. Each order term in each pathway is independent and the covariances for each order add up over all pathways. We first calculate the expectation of the first-order term in a single pathway, which is

$$\sum_{i,i'} \langle f_i x_i f_{i'} x'_{i'} \rangle = \bar{\mu} \sum_{i,i'} \delta_{ii'} x_i x'_{i'} \tag{67}$$

$$= \bar{\mu} g.g' \tag{68}$$

$$= \bar{\mu}(\ell - 2d). \tag{69}$$

Here, we have used $\langle f_i^2 \rangle = \bar{\mu}$, where the $\bar{\mu}$ comes from the probability of locus $i$ being selected for the module as argued previously. The covariance is linear in the distance between genotypes, as one would expect from an additive model. Directly calculating the higher-order terms is more complicated because of the ordering restriction $i > j$ for the second-order term and for higher orders. As noted previously (*Stadler and Happel, 1999*; *Neidhart et al., 2013*; *Agarwala and Fisher, 2019*), these can instead be calculated using combinatorics, which we will demonstrate with the second-order term. We have for the second-order covariance

$$\sum_{i>j,i'>j'} \langle f_{ij} x_i x_j f_{i'j'} x'_{i'} x'_{j'} \rangle = \bar{\mu}^2 \sum_{i>j,i'>j'} \delta_{ii'} \delta_{jj'} x_i x'_{i'} x_j x'_{j'} \tag{70}$$

$$= \bar{\mu}^2 \sum_{i>j} (gg')_i (gg')_j, \tag{71}$$

where the element-wise product $(gg')_i \equiv x_i x'_i$ is 1 if $x_i = x'_i$ match and $-1$ otherwise. If $d$ is the distance between $g$ and $g'$, then the element-wise product $gg'$ has $d$ terms and $\ell - d$ 1 terms. The term in the summation above is 1 if both $(gg')_i$ and $(gg')_j$ are chosen from the $\ell - d$ subset or both are chosen from $d$ subset. This term is $-1$ if one of the two is chosen from the $\ell - d$ subset and the other from the $d$ subset. The number of terms that are 1 s are therefore $\binom{\ell-d}{2}\binom{d}{0} + \binom{\ell-d}{0}\binom{d}{2}$ and the number of terms that are $-1$ s are $\binom{\ell-d}{1}\binom{d}{1}$. This argument is easily extended to higher orders. The general $q$th-order contribution to the covariance is

$$\bar{\mu}^q \sum_{k=0}^{\min(q,d)} (-1)^k \binom{\ell-d}{q-k}\binom{d}{k}. \tag{72}$$

It is easily verified that the first-order term matches. When $d = 0$, we recover the binomial coefficients, as expected. The summation above is precisely the Krawtchouk polynomial $\mathcal{K}_q(d; \ell, 2)$ (*Olver et al., 2010*), which we will denote by $\mathcal{K}_q(\ell, d)$. We therefore have

$$C(d) = \sum_{q=1}^{d} \bar{\mu}^q \mathcal{K}_q(\ell, d). \tag{73}$$

The generating function of the Krawtchouk polynomials yields

$$C(d) = (1+\bar{\mu})^{\ell-d}(1-\bar{\mu})^d - 1. \tag{74}$$

The above expression is consistent with intuition: when $\bar{\mu}\ell \ll 1$, the covariance is linear in $d$, as expected for an additive model. In the opposite limit of $\bar{\mu}\ell \gg 1$ and $0 < d < \ell/2$, the constant term 1 can be ignored, and we see that the covariance is proportional to $e^{-\lambda d}$, where $\lambda = \ln\left(\frac{1+\bar{\mu}}{1-\bar{\mu}}\right)$. Epistasis is maximal when $\bar{\mu} \to 1$, in which case $\lambda \to \infty$ and the covariance rapidly goes to zero with $d$. This is the HoC model of maximal epistasis.

## The landscape of the general CN model

The landscape of the general CN model, where loci can have different $\mu_i$, is no longer stationary but the correlation structure can still be calculated. The $q$th-order contribution to the covariance between genotypes $g$ and $g'$ in the general case is

$$\sum_{i_1 > i_2 > \ldots > i_q} \langle f_{i_1 i_2 \ldots i_q}^2 \rangle \prod_{n=1}^{q} (x_{i_n} x'_{i_n}) = \sum_{i_1 > i_2 > \ldots > i_q} \prod_{n=1}^{q} \mu_{i_n} x_{i_n} x'_{i_n}. \tag{75}$$

It is easy to see that when the contributions from all orders are added up, the covariance $C(g, g')$ has a rather simple product form

$$C(g, g') = \prod_{i=1}^{\ell} (1 + \mu_i x_i x'_i) - 1 \tag{76}$$

The correlation $c(g, g') = C(g, g')/\sqrt{C(g,g)C(g',g')}$ is

$$c(g, g') = \frac{\prod_{i=1}^{\ell}(1 + \mu_i x_i x'_i) - 1}{\prod_{i=1}^{\ell}(1 + \mu_i) - 1}. \tag{77}$$

The above relation is exact. When $\mu_i \ll 1$, $\bar{\mu}\ell \gg 1$, then $1 \pm \mu_i \approx e^{\pm\mu_i}$ and the 1's in the numerator and denominator above can be ignored. We get

$$c(g, g') \approx e^{-2\sum_{i:x_i \neq x'_i} \mu_i}. \tag{78}$$

When $\mu_i$'s are equal, we recover the homogeneous case with exponentially decaying correlations.

## Appendix 3

### Adaptation

The distribution of fitness effects

Long-term adaptation is determined by the DFEs and the stochastic dynamical processes that lead to fixation. Before analyzing the properties of adaptive walks, we clarify what our previous analysis, which ultimately led to *Equation (32)*, means for the DFE.

*Equation (32)* represents the distribution of the fitness effect of a specific mutation at locus $i$ over *all* genotypes in the population that have fitness $y$. We are instead interested in the DFE, where fitness effects are measured for all the mutations of a *particular* genotype that has fitness $y$. The DFE for a particular genotype generally depends on idiosyncratic epistatic interactions between loci. In order to make this explicit, we turn to our analysis framework described previously. For notational convenience, we will assume $\bar{y} = 0$ and $V = 1$ in this subsection and the next.

The DFE over $\ell$ loci for a particular genotype of a fitness $y$ can be thought of as a *sample*, $s_1, s_2, \ldots, s_\ell$, from the conditional joint distribution $P(s_1, s_2, \ldots, s_\ell | y)$. From our assumption of numerous, independent epistatic terms for each locus, this joint distribution is defined entirely in terms of the means and covariances of the $\ell + 1$ variables $y, s_1, s_2, \ldots, s_\ell$. Recall that the fitness effect of a mutation at locus $i$ is $s_i = y_i - y = -2\xi_i$ as defined in *Equation (26)*. As shown previously, we have $\overline{s_i y} = -2v_i$. We also have $\overline{s_i s_j} = 4e_{ij}$ with $e_{ii} = v_i$. The means of all the variables are zero. Based on this covariance structure, we can compute $P(s_1, s_2, \ldots, s_\ell | y)$ using the properties of the conditional distribution of a multivariate normal distribution. It is straightforward to show that conditional on the fitness $y$, the conditional means are given by $\text{Mean}_y(s_i) = -2v_i y$ and the conditional covariances are $\text{Cov}_y(s_i, s_j) = 4(e_{ij} - v_i v_j)$. This relation makes clear that in general the DFE from a sample $s_1, s_2, \ldots, s_\ell \sim P(s_1, s_2, \ldots, s_\ell | y)$ depends on the epistatic interactions between all pairs of loci via $e_{ij}$. Note that $\text{Var}_y(s_i) = 4v_i(1 - v_i)$, which leads to *Equation (32)* for the marginal distribution $P(s_i | y)$ of a particular locus.

The DFE simplifies considerably if we make certain additional assumptions on the nature of epistatic interactions. In particular, the conditional correlation between fitness effects is

$$\text{Corr}_y(s_i, s_j) = \frac{e_{ij} - v_i v_j}{\sqrt{v_i v_j (1 - v_i)(1 - v_j)}}. \tag{79}$$

If we assume the typical VF, $\bar{v}$ is small (i.e., $\bar{v} \ll 1$) and also that $e_{ij}$ is $O(\bar{v}^2)$, then correlations are $O(\bar{v})$ and thus negligible. Then, in a particular sample $s_1, s_2, \ldots, s_\ell$, we can think of each $s_i$ as being drawn independently with mean $-2v_i y$ and variance $4v_i(1 - v_i)$. If $\ell$ is large, then this leads to a DFE

$$\rho(s | y) = \int_0^1 dv (2\sqrt{v(1 - v)})^{-1} P(v) \varphi\left(\frac{s + 2vy}{2\sqrt{v(1 - v)}}\right), \tag{80}$$

where $P(v)$ is the DVFs across the loci and $\varphi$ is the standard normal pdf. Surprisingly, this relation becomes exact for the CN model, where we have shown that $e_{ij} = v_i v_j$ (*Equation (66)*) and therefore the correlations between $s_i$'s vanish. In this case, the DFE is thus determined entirely by the DVF, but we will show later that it has certain universal properties independent of even the DVF. Note that we derive the DFE starting from rather general assumptions on the organization of the genotype-phenotype map in contrast to past models that assume the DFE as a starting point.

Above, we have measured the DFE for the subset of genotypes that have fitness $y$ without regard to their evolutionary history. Over the course of adaptation, mutations are fixed and certain fitness changes are observed. We would then like to measure the DFE for those genotypes that have undergone a particular adaptive trajectory. As we show below, the DFE again simplifies considerably if certain relations hold.

### History dependence

Using an analysis similar to the one in the previous section, we quantify history dependence by calculating the correlations between the new fitness and the adaptive history conditional on the genotypes that have undergone a specific sequence of events in the past.

To be precise, suppose an initial clonal population of fitness $y_0$ gains $k$ successive mutations and the corresponding sequence of fitnesses is $y_1, y_2, \ldots, y_k$. We would like to quantify how the fitness of a new mutation at locus $k+1$, $y_{k+1}$, depends on past fitnesses and the idiosyncratic epistatic interactions between the previous $k$ mutations and the new mutation. The correlation between any two fitnesses $y_i$ and $y_j$ ($i<j$) is given by $1 - 2v_{i+1:j}$, where $v_{i+1:j}$ is the VF of the loci $i+1, i+2, \ldots, j$ (the subscript notation will be used throughout). In general, $v_{i+1:j}$ accounts for the epistatic interactions of all orders between these $j-i$ loci and is expressible in terms of the coefficients of our original complex trait model in *Equation (19)*. One can then write the covariance matrix between $y_0, y_1, \ldots, y_{k+1}$. In block form, this is

$$\Sigma = \begin{pmatrix} \Sigma_{0:k,0:k} & \Sigma_{0:k,k+1} \\ \Sigma_{k+1,0:k} & \Sigma_{k+1,k+1} \end{pmatrix}. \tag{81}$$

The mean of $y_{k+1}$ conditional on $y_0, y_1, \ldots, y_k$ is a linear weighted sum of the past fitnesses:

$$\overline{y_{k+1}} = \Sigma_{k+1,0:k} \Sigma_{0:k,0:k}^{-1} \mathbf{y}, \tag{82}$$

where $y$ is a vector with elements $y_0, y_1, \ldots, y_k$. In other words, $y_{k+1}$ can be written as

$$y_{k+1} = \sum_{i=0}^{k} w_{k+1,i} y_i + \epsilon, \tag{83}$$

where $\epsilon$ is a mean-zero stochastic term that depends on the genotype of the initial population and whose variance can be calculated from Σ. To gain intuition, it is useful to explicitly calculate the case of $k=1$. In this case, the covariance matrix is

$$\Sigma = \begin{pmatrix} 1 & 1-2v_1 & 1-2v_{12} \\ 1-2v_1 & 1 & 1-2v_2 \\ 1-2v_{12} & 1-2v_2 & 1 \end{pmatrix}. \tag{84}$$

We have

$$\Sigma_{2,01} \Sigma_{01,01}^{-1} = \frac{1}{4v_1(1-v_1)} \begin{pmatrix} 1-2v_{12} & 1-2v_1 \end{pmatrix} \begin{pmatrix} 1 & 2v_1-1 \\ 2v_1-1 & 1 \end{pmatrix} \tag{85}$$

$$= \frac{1}{4v_1(1-v_1)} \begin{pmatrix} 1-2v_{12} - (1-2v_1)(1-2v_2) & (1-2v_2) - (1-2v_1)(1-2v_{12}) \end{pmatrix}. \tag{86}$$

We therefore have

$$\overline{y_2} = \frac{v_1 - v_2 + v_{12} - 2v_1v_{12}}{2v_1(1-v_1)} y_1 + \frac{v_1 + v_2 - 2v_1v_2 - v_{12}}{2v_1(1-v_1)} y_0. \tag{87}$$

The dependence on the past has complex dependencies on epistasis between loci 1 and 2 even in this highly simplified case. To identify what contributes to history dependence beyond just the most recent fitness, we rewrite *Equation (87)* as

$$\overline{y_2} = (1-2v_2)y_1 + \frac{(1-2v_1)(v_{12} - v_1 - v_2 + 2v_1v_2)}{2v_1(1-v_1)} y_1 + \frac{v_1 + v_2 - 2v_1v_2 - v_{12}}{2v_1(1-v_1)} y_0. \tag{88}$$

From here, we observe that the dependence on $v_1$ and $y_0$ vanishes precisely when $v_{12} = v_1 + v_2 - 2v_1v_2$. This is not true for all landscapes. However, as noted previously, this is in fact true for the CN model (*Equation (66)*) when $\bar{\mu}\ell \gg 1$. In this case, we get the simple relation

$$y_2 = (1-2v_2)y_1 + 2\sqrt{v_2(1-v_2)}\,\eta, \tag{89}$$

where the pre-factor in the second term comes from normalization and $\eta$ is a Gaussian random variable.

From *Equation (53)*, we have the general relation $v_{12} = v_1 + v_2 - 2e_{12}$, where $e_{12}$ is the epistatic VF between loci 1 and 2. We can then rewrite *Equation (88)* as

$$\overline{y_2} \approx (1 - 2v_2)y_1 + \frac{v_1 v_2 - e_{12}}{v_1} s_1, \tag{90}$$

for $v_1 \ll 1$ and $s_1 = y_1 - y_0$. An intuitive interpretation of this result is presented in the main text.

## Sufficient condition for memoryless fitness gains

The $k = 1$ case suggests that the relation for memoryless fitness gains ($e_{12} = v_1 v_2$) could in fact be true for all $k$ under the CN model, which indeed turns out to be the case, as we show below. Motivated by the $k = 1$ case, we would like to have $w_{k+1,k} = 1 - 2v_{k+1}$ in **Equation (83)** and the rest of the weights equal to zero. If this is the case,

$$y_{k+1} = (1 - 2v_{k+1})y_k + \epsilon. \tag{91}$$

Multiplying both sides by $y_j$ and computing the correlations, we get the condition $1 - 2v_{j+1:k+1} = (1 - 2v_{j+1:k})(1 - 2v_{k+1})$ for all $j$. Therefore, a sufficient condition for memoryless fitness gain is

$$v_{j+1:k+1} = v_{k+1} + v_{j+1:k} - 2v_{k+1}v_{j+1:k} \tag{92}$$

for all $k$ and for all $j<k$. We will now show that this is true for the CN model for $j=0$, that is, $v_{1:k+1} = v_{k+1} + v_{1:k} - 2v_{k+1}v_{1:k}$; the rest trivially follows. Let us first analyze what terms contribute towards $V_{1:k}$ (dividing by $V$ gives $v_{1:k}$). When loci 1 through $k$ are mutated, their effect is to flip the signs of their coefficients in **Equation (19)**. The ones that have changed sign contribute to the decorrelation between the fitnesses before and after the set of mutations, but the epistatic terms that have an even number of flips do not change and therefore their contribution has to be subtracted. $V_{1:k}$ therefore is the sum of squares of all the coefficients in **Equation (19)** whose loci have flipped an odd number of times. To keep track of indices, suppose $i_1, i_2, \ldots$ are used to denote the indices of the $k$ loci (which take values from 1 to $k$) and $j_1, j_2, \ldots$ for the rest. Then,

$$
\begin{aligned}
V_{1:k} &= \sum_{i_1=1}^{k} F_{i_1}^2 + \sum_{i_1 > i_2 > i_3}^{k} F_{i_1 i_2 i_3}^2 + \ldots, \quad \text{where} \\
F_{i_1}^2 &= f_{i_1}^2 + \sum_{j_1 \neq 1:k} f_{i_1 j_1}^2 + \sum_{j_1 > j_2 \neq 1:k} f_{i_1 j_1 j_2}^2 + \ldots, \\
F_{i_1 i_2 i_3}^2 &= f_{i_1 i_2 i_3}^2 + \sum_{j_1 \neq 1:k} f_{i_1 i_2 i_3 j_1}^2 + \sum_{j_1 > j_2 \neq 1:k} f_{i_1 i_2 i_3 j_1 j_2}^2 + \ldots,
\end{aligned}
\tag{93}
$$

and so on. Now, when the $k+1th$ locus is also flipped, to compute $V_{1:k+1}$, we can add up the two variances $V_{1:k}$ and $V_{k+1}$ except for the cross terms that have an even number of sign flips and that include both the $k+1$th locus and the other $k$ loci. These have to be subtracted twice because they appear both in $V_{1:k}$ and $V_{k+1}$. These terms are

$$I_{1:k,k+1} = \sum_{i_1=1}^{k} F_{i_1 k+1}^2 + \sum_{i_1 > i_2 > i_3}^{k} F_{i_1 i_2 i_3 k+1}^2 + \ldots, \tag{94}$$

where the $F$s are defined in a similar fashion as in **Equation (93)** except the sums over $j$s run from $k+2$ to $\ell$ instead of $k+1$ to $\ell$. We get the general relation

$$V_{1:k+1} = V_{k+1} + V_{1:k} - 2I_{1:k,k+1}. \tag{95}$$

For the CN model specifically, we have $\langle f_{i_1 i_2 \ldots i_k}^2 \rangle = \prod_{j=1}^{k} \mu_{i_j}$. This implies

$$\langle I_{1:k,k+1}\rangle = \sum_{i_1=1}^{k}\langle F_{i_1 k+1}^2\rangle + \sum_{i_1 2_3}^{k}\langle F_{i_1 i_2 i_3 k+1}^2\rangle + \dots,$$

$$= \mu_{k+1}\prod_{j=k+2}^{\ell}(1+\mu_j)\left(\sum_{i_1=1}^{k}\mu_{i_1} + \sum_{i_1 2_3}^{k}\mu_{i_1}\mu_{i_2}\mu_{i_3} + \dots\right) \tag{96}$$

$$= \mu_{k+1}\left(\prod_{j=k+2}^{\ell}(1+\mu_j)\right)\left(\prod_{i=1}^{k}(1+\mu_i) - 1\right)$$

Performing a similar calculation for $\langle V_{1:k}\rangle$, we find

$$\langle V_{1:k}\rangle = \prod_{j=k+1}^{\ell}(1+\mu_j)\left(\sum_{i_1=1}^{k}\mu_{i_1} + \sum_{i_1 2_3}^{k}\mu_{i_1}\mu_{i_2}\mu_{i_3} + \dots\right)$$

$$= (1+\mu_{k+1})\left(\prod_{j=k+2}^{\ell}(1+\mu_j)\right)\left(\prod_{i=1}^{k}(1+\mu_i) - 1\right) \tag{97}$$

which gives

$$\langle I_{1:k,k+1}\rangle = v_{k+1}\langle V_{1:k}\rangle. \tag{98}$$

for $\mu\ell \gg 1$ since $v_{k+1} = \mu_{k+1}/(1+\mu_{k+1})$. Dividing *Equation (95)* by $V$ throughout concludes the derivation.

## Adaptedness

Under the conditions of memoryless fitness gains from the previous section, we can write

$$\sigma = -2vz + 2\sqrt{v(1-v)}\nu, \tag{99}$$

where variables have been rescaled as $z = V^{-1/2}(y-\bar{y}), \sigma = V^{-1/2}s, \nu = V^{-1/2}\eta$. This equation suggests various forms for the DFE, $\rho(\sigma|z)$, depending on the DVF, $P(v)$.

$z$ is an intuitive measure of 'adaptedness': (1) when $z$ is negative, the organism is in the unlikely situation of being 'negatively adapted' to the environment. Beneficial mutations are much more likely than deleterious mutations and adaptation is dominated by loci that have a large $v$. (2) When $|z| \ll 1$, the organism is 'neutrally adapted'. The number of beneficial and deleterious mutations is balanced. (3) When $z \gg 1$, the organism is 'well-adapted,' where the DFE is strongly skewed towards deleterious mutations.

We will analyze adaptation in the neutrally adapted and the well-adapted regimes. When the organism is negatively adapted, *Equation (99)* predicts that a few substitutions are sufficient to quickly reach the neutrally adapted state. In addition, we assume the VF of each locus is small, that is, $v \ll 1$ but the number of loci $\ell$ is large enough so that overall epistasis $\approx \bar{v}\ell \gg 1$ ($\bar{v}$ is the mean VF). We can then ignore the $1-v$ factor and rewrite *Equation (99)* as

$$\sigma = -2vz + 2\sqrt{v}\nu. \tag{100}$$

The intuition behind the analytical results is discussed in the main text. We present the formal calculations here.

## Analytical results for an exponential DVF

We begin by calculating the DFE, $\rho(\sigma|z)$, for the specific case when the DVF is an exponential,

$$P(v) = \bar{v}^{-1}e^{-v/\bar{v}}, \tag{101}$$

where $\bar{v} \ll 1$ is the mean VF across loci. The true DVF likely has a large fraction of loci that have no effect; accounting for these loci simply scales the mutation rate and will be ignored in this analysis. The exponential DVF leads to analytical predictions for the shape of the average fitness trajectories under certain simplifying assumptions about the underlying selective forces.

From *Equation (100)*, we have

$$\rho(\sigma|z) = \int\int dv\,d\nu\,P(v)P(\nu)\delta\left(\sigma + 2vz - 2\sqrt{v}\nu\right). \tag{102}$$

Here, $\nu$ is a standard normal random variable, which we integrate out, giving

$$\rho(\sigma|z) = \int_0^1 dv(2\sqrt{v})^{-1}P(v)\varphi\left(\frac{\sigma + 2vz}{2\sqrt{v}}\right), \tag{103}$$

where $\varphi$ is the normal pdf. For $P(v)$ given in *Equation (101)*, this integral can be calculated exactly:

$$\rho(\sigma|z) = \frac{\bar{v}^{-1}}{2\sqrt{2\bar{v}^{-1} + z^2}}e^{-\sigma z/2 - |\sigma|\sqrt{2\bar{v}^{-1} + z^2}/2}. \tag{104}$$

The resulting DFE is a double exponential with scale $\sim(z/2 \pm \sqrt{2\bar{v}^{-1} + z^2}/2)^{-1}$. For $|z| \ll 1$ small, the DFE is symmetric around the origin, as expected, with a scale determined by the DVF. As $z$ increases, the DFE skews towards deleterious effects. The typical magnitudes of beneficial and deleterious effects are not independent from the overall ratio of beneficial to deleterious mutations. The well-adapted regime is reached when $z^2$ is comparable to $\bar{v}^{-1}$. To clearly delineate the two regimes, it is useful to define new variables $x = z\sqrt{\bar{v}/2}$ and $\lambda = \sigma\sqrt{2/\bar{v}}$. In the new variables, the DFE is

$$\rho(\lambda|x) = \frac{1}{4\sqrt{1 + x^2}}e^{-\frac{\lambda x + |\lambda|\sqrt{1 + x^2}}{2}}. \tag{105}$$

The neutrally adapted and well-adapted regimes then correspond to $x \lesssim 1$ and $x \gtrsim 1$, respectively. The mean rate of adaptation in units of generations on the $x$ scale is $\langle dx \rangle = \langle dz \rangle\sqrt{\bar{v}/2} = \langle \sigma \rangle\sqrt{\bar{v}/2} = \langle \lambda \rangle\bar{v}/2$, where the expectation is taken over fixation probabilities and the DFE. We assume SSWM so that $p_{\text{fix}}(\sigma) \sim 2\sigma$ or $p_{\text{fix}}(\lambda) \sim 2\lambda\sqrt{\bar{v}/2}$ for positive $\lambda$ and 0 otherwise. We find

$$\langle dx \rangle = NU\bar{v}/2\int_{-\infty}^\infty d\lambda\, p_{\text{fix}}(\lambda)\rho(\lambda|x)\lambda \tag{106}$$

$$= NU\bar{v}\sqrt{\bar{v}/2}\int_0^\infty d\lambda\,\lambda^2\rho(\lambda|x), \tag{107}$$

where $N$ is population size, and $U$ is the effective mutation rate for loci with nonzero effect. Integrating over $\lambda$, we get

$$\langle dx \rangle = \frac{NU(2\bar{v})^{3/2}}{\sqrt{1 + x^2}(x + \sqrt{1 + x^2})^3}. \tag{108}$$

Integrating over $dx$ starting from an initial $x_0$, we obtain an approximation to the mean fitness trajectory

$$NU(2\bar{v})^{3/2}n_{\text{gen}} = \int_{x_0}^x dx\sqrt{1 + x^2}(x + \sqrt{1 + x^2})^3 = T(x) - T(x_0), \tag{109}$$

where $T(x) = \frac{4x^5}{5} + \frac{5x^3}{3} + \frac{\sqrt{x^2 + 1}}{15}\left(12x^4 + 19x^2 + 7\right) + x.$ \tag{110}

Note that this is only an approximation for small $x$ since the typical fixed beneficial effect is a discrete jump. We show the result from *Equation (109)* in *Appendix 1—figure 1a*, where the dependencies on the equilibrium fitness $\bar{y}$ and genetic variance $V$ are highlighted. There are two independent parameters that determine the scale ($\sqrt{\bar{v}/2V}$) and location ($\bar{y}$) of the fitness, and one parameter ($NUn_{\text{gen}}$) that determines the time scale. From *Equation (109)*, the $x \sim t$ and $x \sim t^{1/5}$ scalings in the neutrally adapted ($x \ll 1$) and well-adapted regime ($x \gg 1$) respectively are apparent.

The number of substitutions, $n_s$, as a function of $x$ can also be calculated under the SSWM assumption. We get

$$\bar{v}n_s = N(x) - N(x_0) \tag{111}$$

$$\text{where} \quad N(x) = x(x + \sqrt{x^2+1})/4 + \sinh^{-1}(x)/4, \tag{112}$$

which can be mapped onto time using *Equation (109)*. The scalings are $n_s \sim x \sim t$ in the neutrally adapted regime and $n_s \sim x^2 \sim t^{2/5}$ in the well-adapted regime.

## Asymptotics and general scaling results

We now derive the asymptotic properties of the DFE in the well-adapted regime ($\bar{v}^{1/2}z \gg 1$). Writing out the Gaussian pdf in *Equation (80)*, we have

$$\rho(\sigma|z) = \int_0^1 dv (2\sqrt{2\pi v})^{-1} P(v) e^{-\frac{(\sigma+2vz)^2}{8v}} \tag{113}$$

$$= e^{-\sigma z/2} \int_0^1 dv (2\sqrt{2\pi v})^{-1} P(v) e^{-\frac{\sigma^2}{8v} - \frac{vz^2}{2}} \tag{114}$$

$$= e^{-\sigma z/2} \int_0^1 du (2\sqrt{2\pi u y})^{-1} P(u/z) e^{-\frac{z}{2}\left(\frac{\sigma^2}{4u} + u\right)}, \tag{115}$$

where the change of variables $v = u/z$ is used. When $z \gg 1$, Laplace's method can be used. The exponent is minimized when $u = |\sigma|/2$, which gives

$$\rho(\sigma|z) \approx (2z)^{-1} P(|\sigma|/2z) e^{-\frac{z}{2}(\sigma + |\sigma|)}. \tag{116}$$

The contribution towards a fitness effect $\sigma$ at large $z$ comes largely from loci with VF $v \approx |\sigma|/2z$. The exponential form of the beneficial DFE is determined entirely by the Gaussian tails of the genotype-dependent term. The argument can be easily generalized to non-Gaussian tail probabilities using a similar calculation. *Equation (116)* implies that the ratio of the probabilities of beneficial and deleterious mutations is independent of the DVF as long as it has sufficient mass at $v = |\sigma|/2z$:

$$\frac{\rho(\sigma|z)}{\rho(-\sigma|z)} = e^{-\sigma z}. \tag{117}$$

Such a relationship has been hypothesized previously based on simulations of an additive finite-sites model and the form of $p_{\text{fix}}$ close to the high-interference limit that could result in a fitness plateau (*Rice et al., 2015*). Using our theory, we have shown that this result is indeed true independent of the DVF and under our core hypotheses of normality and memoryless fitness gains. If $p_{\text{fix}} \sim e^{\sqrt{V}T_c\sigma}$ in the high-interference limit, where $T_c$ is the coalescent time, fitness should plateau when $p_{\text{fix}}(\sigma)\rho(\sigma|z) = p_{\text{fix}}(-\sigma)\rho(-\sigma|z)$, which is at

$$z_{\text{plateau}} = 2\sqrt{V}T_c. \tag{118}$$

The $\sqrt{V}$ appears to account for the rescaling $\sigma = s/\sqrt{V}$.

We get *Equation (116)* only if $P(v)$ has probability mass at $v = |\sigma|/2z$. This is not the case in the exponentially correlated fitness landscape model with homogeneous loci, that is, if, for instance, $P(v) = \delta(v - \bar{v})$, we instead get from *Equation (80)*

$$\rho(\sigma|z) = \varphi\left(\frac{\sigma + 2\bar{v}z}{2\sqrt{\bar{v}}}\right). \tag{119}$$

The DFE in this case is therefore a Gaussian with mean shifting towards the deleterious side. The ratio of beneficial to deleterious mutations goes rapidly to zero as $\simeq \varphi(\sqrt{\bar{v}}z)/\sqrt{\bar{v}}z$ and adaptation sharply plateaus beyond $z = \bar{v}^{-1/2}$.

From *Equation (116)*, various scaling results can be derived that apply independent of $P(v)$ and the Gaussian assumption on $v$. We retain the exponential form for convenience. The rate of beneficial mutations is

$$U_b = U \int_0^\infty d\sigma (2z)^{-1} P(|\sigma|/2z) e^{-\sigma z} \tag{120}$$

$$= \frac{U}{2z^2} \int_0^\infty dw P(w/2z^2) e^{-w}. \tag{121}$$

The integral has an effective upper cutoff at $w \sim O(1)$ and can be approximated as $U_b \approx U P(0)/2z^2 \sim z^{-2}$ under certain assumptions for $P(v)$ for small $v$. The pre-factor $P(0)$ suggests that the number of beneficial mutations depends on the number of small-effect loci. While strong epistatic loci drive adaptation in the neutrally adapted regime, adaptation in the well-adapted regime is instead driven by weakly epistatic loci.

From *Equation (116)*, the typical size of a beneficial mutation is $\sigma \sim z^{-1}$. Under SSWM, $p_{\text{fix}} \sim \sigma \sim z^{-1}$. As argued previously, since $U_b \sim z^{-2}$, we get $dz/dt \sim z^{-4}$ and therefore we obtain the two scaling relations

$$z \sim t^{1/5}, n_s \sim t^{2/5}, \tag{122}$$

which apply independently of the form of the DVF in the well-adapted regime.

