## [Decision Letter]

Thank you for submitting your article "Global epistasis emerges from a generic model of a complex trait" for consideration by *eLife*. Your article has been reviewed by 3 peer reviewers, and the evaluation has been overseen by Naama Barkai as the Senior and Reviewing Editor. The following individuals involved in review of your submission have agreed to reveal their identity: Dmitri A Petrov (Reviewer #1); Kristina Crona (Reviewer #2); Joachim Krug (Reviewer #3).

As you will see below, all reviewers were excited about the quality of the work. Still, to make is accessible to broad audience, the paper would benefit from revising the writing, making it more accessible and precise. Please relate to all comments below.

Reviewer #1 (Recommendations for the authors):

I have little to add to the paper and I strongly support its publication.

Some points that come to mind as a reader:

1. I would have liked more detailed analysis of the connection between the extreme value theory and the results in the paper.

2. most experimental evolution is not done in the WMSS regime and indeed the authors suggest that the failure of their predictions in the cases of LTEE might be due to the strong mutation regime of the LTEE. Can more be said about the expected differences in predictions in the strong mutation regime?

3. I imagine these arguments were made in engineering or catastrophe theory or economics or control theory – is this true? The results feel so generic that they would apply to all complex systems.

4. What do you results say about the evolution of modularity and how epistasis would evolve?

Reviewer #2 (Recommendations for the authors):

The authors should address the fact that diminishing return and increases costs occur for landscapes beyond the type considered. For instance, consider a biallelic 20-locus fitness landscapes associated with stabilizing selection, where genotypes are represented as bit strings and

wg=1-∣∑gi-10∣20

A mutation 0 ↦ 1 for a genotype *g* results in the fitness difference

-120if∑gi≥10120if∑gi<10

There is no regression effect here. However, by adding random noise to the fitness values and instead assume that wg=1-∣∑gi-10∣20+εg the result is a landscape with both diminishing return and increased costs effects.

A. Major comments and recurrent issues.

1. Throughout the paper, the manuscript seems to assume SSWM, and fall back on Gillespie’s theory for small fitness differences between genotypes. For instance the author writes (Page 15):”This analysis suggests that during adaptation, since selection favors mutations with stronger fitness effects on the current background, a mutation that interacts less with previous mutations is more likely to be selected.” However, if two beneficial mutations have strong effects than they are about equally likely to go to fixation.

2. Every mathematical expression in the main text should have an explanation or a reference (for biology journals that is a reasonable requirement).

3. References are missing throughout the manuscript. There is not even a reference for”Krawtchouk polynomials”.

4. The SI is too difficult to read in my view. More detailed argument would help, and also useful references for readers who need to catch up, or review, say Fourier representations or Brownian motion.

5. Throughout the main text, the manuscript abuses notation for Fourier coefficients (without comments). The expansion on Page 2 assumes that *i > j* for *f_ij_*(and similarly) but later on the authors use *f_ij_*where *i < j* (at first the notation was confusing to me, especially subscripts of the type *i* 6 = *j > k*). At least a comment should be provided, or otherwise the notation should be changed (in fact the manuscript uses different notation in the SI, Page 9).

6. The following type of expression was unclear to me before I read the SI (on Page 6 and other places in the manuscript):

∑ *f_j_ f_ij_*+∑ *f_jk_ f_ijk_*+*….*

7. The expression”slope for a double mutant”, and similarly, sounds strange (on Page 9 and other places in the manuscript). Here the fitness of a double mutant is compared to the sum of the fitnesses for the corresponding single mutants (something similar should be stated for clarity).

8. It would be great with an explicit example (a toy example) of a landscape where global epistasis varies substantially between loci, for some intuitive understanding.

B. Detailed comments in order of appearance. Most comments below are very minor and concerns notation or wording. The list is not intended to be complete (in fact most comments concern the first few pages and the corresponding part of the SI). Similar minor issues were noted throughout the manuscript, but all of it can be handled swiftly.

1. Page 1:” However, the mechanistic basis of this global epistasis remains unknown.”

The manuscript provides statistical rather than mechanistic explanations. (A mechanistic explanation would concern protein folding or something.)

2. Page 5: “In other words, Equation 2 implies that widespread independent idiosyncratic epistatic interactions lead to the observed patterns of diminishing-returns and increasing-costs epistasis.”

Equation 2 does not imply anything, since such an expansion is possible for any genotype. It would make more sense to simply say:”We argue that wide spread independent idiosyncratic epistatic lead to diminishing return and increasing costs epistasis.” Alternatively, explain the connection to Equation 2 in a more precise way.

(The heuristic argument on the same page is very nice!)

3. Page 6, L 103: The definition of v~iis very important and should be clarified. Explain the symbols ∑ *f_j_ f_ij_*+∑ *f_jk_ f_ijk_*+*…* and explain the symbols in the denominator. (It would also be nice with a brief description of how the expression is composed from the regression argument in the SI.)

4. Page 6, L108:”Note that these results hold for any fitness landscape”. This is not true, as the text also states a few lines later.

5. Page 6, L 120: Why does the sum scale as l?

6. Page 8, L 162:”The theory additionally predicts.…” What exactly is the claim? Does the claim concern any double mutant such that the corresponding single mutations are beneficial?

7. Page 8, L 171:”To test our analytical results” Sounds strange, simulations do not verify analytical results. What exactly is the purpose of the simulation (an illustration of the analytical results perhaps)?

8. SI: It would be much easier to read the SI if it started with Part A followed by Part I. The reason is that Part I uses *v_i_*that is defined in Part A.

9. SI: The notation makes the reading more difficult than necessary in a few places. For instance the symbols *y_i_*and *y_i_*are unrelated, whereas *y_i_*and *ξ_i_*are closely related. In addition, more informative notation could save time for a reader, such as *y(x_i_*= 1)*, y(x_i_*= −1) rather than y~iand y^i.

10. SI, L102: Since v~iis important, it seems reasonable with more explanations of the derivation. Properties of the Fourier expansions are used for cancelling out terms and other simplifications (which is also mentioned), but explicit arguments are not provided.

11. Why is v~idefined as half of the negative slope? Later on -2v~ishows up in formulas. It would make more sense to define v~ias the slope, which would also make it easier to interpret some formulas in the main text.

Reviewer #3 (Recommendations for the authors):

The results for the symmetric model are much simpler and more transparent – the slope of the regression is strictly negative, and has a natural interpretation in terms of variance fraction, whereas the corresponding expression (4) in the directed case looks intimidatingly complicated and obscure. I had a hard time understanding the difference between the two cases and the meaning of the statement that in the symmetric model "the fitness effects of both *x_i_* = -1 \to +1 and *x_i_* = +1 \to -1 are regressed against the background fitness". My interpretation of this is now the following:

To apply the symmetric model to experimental data, along with each mutation also the corresponding reversion would have to be included in the data set. Whereas the selection coefficient of the reversion is simply the negative of that of the mutation, it happens on a different background, and therefore the effect of including it in the regression is not trivial. In practice, including the reversions is of course not possible, and therefore the directed model is needed, although conceptually the symmetric model is more appealing.

If the authors agree with the above, I suggest that they add some explanation along these lines in the manuscript.

The following points concern the same issue:

1. In the application to experimental data, it is of course also not possible (or meaningful) to distinguish between the two directions *x_i_* = -1 \to +1 and *x_i_* = +1 \to -1. It is therefore a bit worrisome that (3,4) depend on the direction. I assume that in practice the data analysis is anyway restricted to the WE limit where the difference between the two direction does not matter. Is that correct? If so, it should be explicitly stated.

2. The description of how the model is applied to the data of Johnson et al. (Methods and Materials, lines 524 ff.) is not entirely clear. Why can the variance fraction be estimated from the Pearson correlation between *y_i_* and *y*? And what does it mean (in line 534) that the slope of the regression is either 2*v_i_* – 1 or 2v~i -1? If the directed model applies, the slope should always be 2v~i – 1. Related to this, in Figure 3a it seems -2v should be replaced by -2v~.

---

## [Author Response]

Reviewer #1 (Recommendations for the authors):I have little to add to the paper and I strongly support its publication.Some points that come to mind as a reader:1. I would have liked more detailed analysis of the connection between the extreme value theory and the results in the paper.

We have now included a derivation of Orr’s result and how that relates to the result we obtain (lines 406-414 of RM). Our results are consistent with Orr’s result, and give additional information about how the beneficial DFE and the rate of beneficial mutations change with adaptedness.

2. most experimental evolution is not done in the WMSS regime and indeed the authors suggest that the failure of their predictions in the cases of LTEE might be due to the strong mutation regime of the LTEE. Can more be said about the expected differences in predictions in the strong mutation regime?

This is an interesting issue, and we have added a few comments about the dynamics in the strong mutation regime (lines 523-530 in the RM). Future work will be required to say anything more quantitative, because while existing theory has extensively analyzed this regime for a fixed DFE and weak epistasis, there is much less understanding of how strong mutations interact with global epistasis.

3. I imagine these arguments were made in engineering or catastrophe theory or economics or control theory – is this true? The results feel so generic that they would apply to all complex systems.

We wondered the same and are unaware of a similar result in other fields. Perhaps the most closely related result is the fluctuation-dissipation theorem from statistical physics, which is also based on the idea that many independent interactions lead to a negative linear feedback and Gaussian fluctuations of proportional magnitudes. We discuss this relationship in the SI (see section on Relationship to fluctuation-dissipation theorem).

4. What do you results say about the evolution of modularity and how epistasis would evolve?

The Fourier framework here assumes a fixed landscape and does not capture global architectural changes in the landscape over the course of long-term evolution. We think the evolution of modularity is an interesting question and plan to develop a framework that incorporates both epistasis and a plastic landscape in follow-up work. Our intuition is that modularity is typically favored in changing environments, as expected. However, our results suggest that a fully-connected architecture has on average mutations of large effect due to epistasis, which can speed adaptation in the neutrally-adapted regime. We suspect that a non-modular architecture may be favored in certain cases depending on how much environments differ and change.

Reviewer #2 (Recommendations for the authors):The authors should address the fact that diminishing return and increases costs occur for landscapes beyond the type considered. For instance, consider a biallelic 20-locus fitness landscapes associated with stabilizing selection, where genotypes are represented as bit strings and

wg=1-∣∑gi-10∣20

A mutation 0 ↦ 1 for a genotype g results in the fitness difference

-120if∑gi≥10120if∑gi<10

There is no regression effect here. However, by adding random noise to the fitness values and instead assume that wg=1-∣∑gi-10∣20+εg the result is a landscape with both diminishing return and increased costs effects.

A. Major comments and recurrent issues.1. Throughout the paper, the manuscript seems to assume SSWM, and fall back on Gillespie’s theory for small fitness differences between genotypes. For instance the author writes (Page 15):”This analysis suggests that during adaptation, since selection favors mutations with stronger fitness effects on the current background, a mutation that interacts less with previous mutations is more likely to be selected.” However, if two beneficial mutations have strong effects than they are about equally likely to go to fixation.

We have expanded on our SSWM and small fitness differences assumptions, and included the expression for the fixation probability (line 373-377 in RM). We expect the small differences assumption to have minimal effect on the long-term dynamics as the typical beneficial fitness effect in the well-adapted regime is small relative to the fitness (~ 1/2*z*^2^, where z is adaptedness and z >> 1 in the well-adapted regime). We have also added a brief comment in the Discussion regarding non-SSWM dynamics.

2. Every mathematical expression in the main text should have an explanation or a reference (for biology journals that is a reasonable requirement).

We have added references, referred to the SI or elaborated on the derivation for each of the numbered equations in the main text.

3. References are missing throughout the manuscript. There is not even a reference for”Krawtchouk polynomials”.

We have added a reference for Krawtchouk polynomials and elsewhere (line 929).

4. The SI is too difficult to read in my view. More detailed argument would help, and also useful references for readers who need to catch up, or review, say Fourier representations or Brownian motion.

We have added references to Brownian motion (lines 845-848) and Fourier representations (lines 732).

5. Throughout the main text, the manuscript abuses notation for Fourier coefficients (without comments). The expansion on Page 2 assumes that i > j for f_ij_ (and similarly) but later on the authors use f_ij_ where i < j (at first the notation was confusing to me, especially subscripts of the type i 6 = j > k). At least a comment should be provided, or otherwise the notation should be changed (in fact the manuscript uses different notation in the SI, Page 9).

We have added a footnote before Equation 4 explaining the notation.

6. The following type of expression was unclear to me before I read the SI (on Page 6 and other places in the manuscript):∑ f_j_ f_ij_ +∑ f_jk_ f_ijk_ +….

The symbols are now explained in a footnote (point 5 above). We have added a sentence to hint where v~ is coming from (line 118-120 of RM).

7. The expression”slope for a double mutant”, and similarly, sounds strange (on Page 9 and other places in the manuscript). Here the fitness of a double mutant is compared to the sum of the fitnesses for the corresponding single mutants (something similar should be stated for clarity).

We agree and have modified this phrase for clarity (lines 183-185 and 210-211 of RM).

8. It would be great with an explicit example (a toy example) of a landscape where global epistasis varies substantially between loci, for some intuitive understanding.

We note that the CN model is a landscape where global epistasis varies across loci. We have added a sentence to highlight that the CN model serves this purpose (lines 322-324 in RM) As a side note, this was our original motivation for constructing the CN model. Coincidentally, it turned out the CN model satisfied the minimal contingency condition as well.

B. Detailed comments in order of appearance. Most comments below are very minor and concerns notation or wording. The list is not intended to be complete (in fact most comments concern the first few pages and the corresponding part of the SI). Similar minor issues were noted throughout the manuscript, but all of it can be handled swiftly.1. Page 1:” However, the mechanistic basis of this global epistasis remains unknown.”The manuscript provides statistical rather than mechanistic explanations. (A mechanistic explanation would concern protein folding or something.)

We agree and have removed “mechanistic” here and elsewhere.

2. Page 5:”In other words, Equation 2 implies that widespread independent idiosyncratic epistatic interactions lead to the observed patterns of diminishing-returns and increasing-costs epistasis.”Equation 2 does not imply anything, since such an expansion is possible for any genotype. It would make more sense to simply say:”We argue that wide spread independent idiosyncratic epistatic lead to diminishing return and increasing costs epistasis.” Alternatively, explain the connection to Equation 2 in a more precise way.(The heuristic argument on the same page is very nice!)

We agree and have removed “Equation 2 implies that”.

3. Page 6, L 103: The definition of v~i*is very important and should be clarified. Explain the symbols ∑ f_j_ f_ij_ +∑ f_jk_ f_ijk_ +… and explain the symbols in the denominator. (It would also be nice with a brief description of how the expression is composed from the regression argument in the SI.)*

The changes are described in Point 6 of major points above.

4. Page 6, L108:”Note that these results hold for any fitness landscape”. This is not true, as the text also states a few lines later.

We agree that this statement can confuse readers and have removed it.

*5. Page 6, L 120: Why does the sum scale as* l*?*

The scaling with l follows from a central limit argument, as we are adding up l independent terms with arbitrary signs.

6. Page 8, L 162:”The theory additionally predicts.…” What exactly is the claim? Does the claim concern any double mutant such that the corresponding single mutations are beneficial?

We have modified this (see point 7 of major points above).

7. Page 8, L 171:”To test our analytical results” Sounds strange, simulations do not verify analytical results. What exactly is the purpose of the simulation (an illustration of the analytical results perhaps)?

We agree and have changed “test” to “illustrate”.

8. SI: It would be much easier to read the SI if it started with Part A followed by Part I. The reason is that Part I uses v_i_ that is defined in Part A.

We agree that the *v_i_* is defined later in part A, but we feel that part A won’t be motivated well if it’s presented first. For this reason, we chose to keep the existing order and have highlighted further that *v_i_* will be defined later.

9. SI: The notation makes the reading more difficult than necessary in a few places. For instance the symbols y_i_ and y_i_ are unrelated, whereas y_i_ and ξ_i_ are closely related. In addition, more informative notation could save time for a reader, such as y(x_i_ = 1), y(x_i_ = −1) rather than y~i*and* y^i.

We agree and have changed the notation for these quantities.

10. SI, L102: Since v~i*is important, it seems reasonable with more explanations of the derivation. Properties of the Fourier expansions are used for cancelling out terms and other simplifications (which is also mentioned), but explicit arguments are not provided.*

We have elaborated further on how this calculation is done (line 779 of RM).

11. Why is v~i*defined as half of the negative slope? Later on* -2v~i*shows up in formulas. It would make more sense to define* v~i*as the slope, which would also make it easier to interpret some formulas in the main text.*

*v_i_* has an interpretation as the variance fraction and additionally, for the CN model we show that *v_i_* is the fraction of pathways involving locus i. For these reasons, we think it is more convenient to refer to *v_i_*, which is half the negative slope, rather than the slope. We have added a sentence in the text explaining our choice (lines 170-172 in RM).

Reviewer #3 (Recommendations for the authors):The results for the symmetric model are much simpler and more transparent – the slope of the regression is strictly negative, and has a natural interpretation in terms of variance fraction, whereas the corresponding expression (4) in the directed case looks intimidatingly complicated and obscure. I had a hard time understanding the difference between the two cases and the meaning of the statement that in the symmetric model "the fitness effects of both x_i_ = -1 \to +1 and x_i_ = +1 \to -1 are regressed against the background fitness". My interpretation of this is now the following:To apply the symmetric model to experimental data, along with each mutation also the corresponding reversion would have to be included in the data set. Whereas the selection coefficient of the reversion is simply the negative of that of the mutation, it happens on a different background, and therefore the effect of including it in the regression is not trivial. In practice, including the reversions is of course not possible, and therefore the directed model is needed, although conceptually the symmetric model is more appealing.If the authors agree with the above, I suggest that they add some explanation along these lines in the manuscript.

We agree with this interpretation and that the symmetric model is conceptually more appealing. We chose to include the directed mutation case as it is closer to how experimental data is often presented in practice (i.e., Figure 1c,d and Equation 1). We modified and added a few sentences (lines 162-165, 174-176 in RM) to make this clearer.

The following points concern the same issue:1. In the application to experimental data, it is of course also not possible (or meaningful) to distinguish between the two directions x_i_ = -1 \to +1 and x_i_ = +1 \to -1. It is therefore a bit worrisome that (3,4) depend on the direction. I assume that in practice the data analysis is anyway restricted to the WE limit where the difference between the two direction does not matter. Is that correct? If so, it should be explicitly stated.

The sign convention indeed shouldn’t matter, irrespective of whether the landscape is in the WE limit. We modified and added a few sentences to make this clear. If the convention is reversed, we can check that the value of v~i remains the same. This is because the odd-order Fourier coefficients also change sign (lines 120-124 of RM).

2. The description of how the model is applied to the data of Johnson et al. (Methods and Materials, lines 524 ff.) is not entirely clear. Why can the variance fraction be estimated from the Pearson correlation between y_i_ and y? And what does it mean (in line 534) that the slope of the regression is either 2v_i_ – 1 or 2v~i *-1? If the directed model applies, the slope should always be* -2v~i *– 1. Related to this, in Figure 3a it seems -2v should be replaced by* -2v~.

We thank the reviewer for spotting these errors (note that this is a misprint and the values of *v_i_* shown in the Figures were computed correctly). The variance fraction is the Pearson correlation when data from both +1 -> -1 and -1 > +1 is included. We have modified to address the three points above (lines 577-579 and 582 of RM).